# Recent decrease of the impact of tropical temperature on the carbon cycle linked to increased precipitation

Wenmin Zhang [1] ✉, Guy Schurgers [1], Josep Peñuelas [2,3], Rasmus Fensholt [1], Hui Yang[4], Jing Tang [5,6], Xiaowei Tong [7], Philippe Ciais[8] & Martin Brandt [1]

The atmospheric $CO_2$ growth rate (CGR) variability is largely controlled by tropical temperature fluctuations. The sensitivity of CGR to tropical temperature $(\gamma_{CGR}^T)$ has strongly increased since 1960, but here we show that this trend has ceased. Here, we use the long-term $CO_2$ records from Mauna Loa and the South Pole to compute CGR, and show that $\gamma_{CGR}^T$ increased by 200% from 1960–1979 to 1979–2000 but then decreased by 117% from 1980–2001 to 2001–2020, almost returning back to the level of the 1960s. Variations in $\gamma_{CGR}^T$ are significantly correlated with changes in precipitation at a bi-decadal scale. These findings are further corroborated by results from a dynamic vegetation model, collectively suggesting that increases in precipitation control the decreased $\gamma_{CGR}^T$ during recent decades. Our results indicate that wetter conditions have led to a decoupling of the impact of the tropical temperature variation on the carbon cycle.

Increased atmospheric concentrations of greenhouse gases have caused global warming that profoundly influences terrestrial ecosystems and the global carbon uptake, which in turn reinforces climate change[1–5]. Increased atmospheric $CO_2$ concentrations also have a positive effect on photosynthesis[6], promoting vegetation growth[7]. Consequently, an increase in the net terrestrial carbon sink has been documented for the last 50 years[8], which in turn has contributed to slowing down the ongoing global warming[3]. These processes constitute the interactions of climate and carbon cycle (Supplementary Fig. 1) that determine future climate change, and particularly the carbon uptake of the tropics has been documented to play a critical role in regulating the carbon cycle[1]. However, capturing the sensitivity of tropical carbon sequestration to climate change remains a challenging task for Earth System Models (ESMs) when predicting future climate change[1,3,5].

Interannual variations of the atmospheric $CO_2$ growth rate (CGR) are tightly coupled to the variations in tropical air temperature[5,9,10], which are themselves strongly associated with the El Niño Southern Oscillation (ENSO)[10]. The sensitivity of CGR to tropical temperature $(\gamma_{CGR}^T)$ has been reported to have increased since 1960 and was suggested to be controlled by emerging water stress[9]. A recent study documented that an increase in tropical extreme droughts could also amplify the CGR variability[11]. Moreover, a continuous increase in temperature that may approach or exceed an optimal temperature of plant photosynthesis[12] may have negative effects on the net ecosystem carbon sink[13], even though tropical forests may acclimate to a warmer climate[14–16]. In this case, a very warm year at present time could be more detrimental for photosynthesis than the same anomaly happening decades ago. Multiple lines of evidence indicate that the net terrestrial carbon uptake has saturated or decreased over tropical

[1]Department of Geosciences and Natural Resource Management, University of Copenhagen, Copenhagen, Denmark. [2]CSIC, Global Ecology Unit CREAF-CEAB-UAB, Cerdanyola del Vallès, 08193 Catalonia, Spain. [3]CREAF, Cerdanyola del Vallès, 08193 Catalonia, Spain. [4]Department for Biogeochemical Integration, Max-Planck-Institute for Biogeochemistry, 07745 Jena, Germany. [5]Department of Physical Geography and Ecosyste m Science, Lund University, Lund, Sweden. [6]Department of Biology, University of Copenhagen, Copenhagen, Denmark. [7]Key Laboratory for Agro-ecological Processes in Subtropical Region, Institute of Subtropical Agriculture, Chinese Academy of Sciences, Changsha 410125, China. [8]Laboratoire des Sciences du Climat et de l'Environnement, CEA CNRS UVSQ, Gif-sur-Yvette, France. ✉e-mail: wenminzhg@gmail.com

areas in recent decades[17–19]. This is caused by increased frequency of droughts, land degradation, deforestation, cropland expansion and changes in forest wildfires[4,19–21]. Collectively, these changes are expected to influence the established impact of temperature on carbon fluxes in the tropics, yet the primary factor controlling the trends of this effect remains unclear.

Here, we investigate the changes in $\gamma_{CGR}^{T}$ using the longest $CO_2$ records from Mauna Loa (1960–2020) and the South Pole (1980–2020) atmospheric stations, and climatic data from CRU-4 and ERA5. We subsequently analyze the variations in $\gamma_{CGR}^{T}$ under different water stress conditions and used a dynamic global vegetation model (LPJ-GUESS) to separate the contributions of changes in climate variables, atmospheric $CO_2$ concentrations, nitrogen deposition and land cover change on the changes in $\gamma_{NBP}^{T}$ (NBP, net biome productivity). Finally, we assess variations in $\gamma_{NBP}^{T}$ simulated by 33 ESMs in the Coupled Model Intercomparison Project Phase 6 (CMIP6), given their state-of-the-art predictions of climate change.

## Results

We re-examine the relationship between CGR and temperature, precipitation and solar radiation, the latter using cloud cover as a proxy, during 1960–2020. This analysis is performed using global and tropical climate variables (24 °N to 24 °S) in a partial regression analysis. The results indicate that the correlation of CGR with tropical temperature is stronger than with precipitation and solar radiation (with an average correlation coefficient of 0.31) in the tropical regions (Supplementary Fig. 2). Temperature is then averaged over the tropics to determine the interannual $\gamma_{CGR}^{T}$ (Fig. 1a). Changes in $\gamma_{CGR}^{T}$ are studied using a 20-year moving window over 1960–2020 (Fig. 1b and Supplementary Fig. 3). We exclude the years of 1992 and 1993 following the Mount Pinatubo eruption due to their significant impact on CGR by increased diffuse light[22]. Significant positive partial correlations between variations in CGR and temperature are identified in each moving window with an average $r$ of 0.58 ($P < 0.05$) (Supplementary Figs. 4 and 5), while the direction and magnitude of the partial correlations of CGR with precipitation and solar radiation vary greatly over time. Multiple regressions between CGR and climate variables explain almost half of the variations averaged over all moving windows (Supplementary Fig. 6).

To further study the changes in $\gamma_{CGR}^{T}$, we split the study period into two periods of equal length: 1960–2000 and 1980–2020. We find that $\gamma_{CGR}^{T}$ increased by 200% from 1960–1979 (1.64 ± 0.54 PgC y⁻¹ K⁻¹) to 1979–2000 (4.93 ± 1.66 PgC y⁻¹ K⁻¹), corresponding to a trend of 0.19 ± 0.03 PgC y⁻¹ K⁻¹ per year. However, for the second period we observe a negative trend of 0.13 ± 0.02 PgC y⁻¹ K⁻¹ per year and an overall decrease of 117% (2001–2020: 2.27 ± 1.19 PgC y⁻¹ K⁻¹), which brings the values of $\gamma_{CGR}^{T}$ down to a similar level as observed in the 1960s (Fig. 1b, c). The decreased $\gamma_{CGR}^{T}$ for the last 40 years is also found when using $CO_2$ data from the South Pole station (Supplementary Fig. 7) and when applying different temperature data (ERA5, Supplementary Fig. 3). The significantly ($P < 0.05$) increasing and decreasing trends are identified with a specific Mann-Kendall trend test[23] accounting for serial autocorrelations in $\gamma_{CGR}^{T}$ (Fig. 1c). Changes in $\gamma_{CGR}^{T}$ at a high temporal frequency (where monthly climate variables and CGR are calculated using a 12-month moving window) also follow the observed pattern of changes in annual $\gamma_{CGR}^{T}$ (Supplementary Fig. 3c, d).

The variations in $\gamma_{CGR}^{T}$ are assumed to be regulated by moisture conditions[9,11], so we use the standardized deviation ($\sigma$) of the long-term detrended precipitation, the self-calibrating Palmer Drought Severity Index (scPDSI) and the Standardized Precipitation and Evaporation Index (SPEI) to divide the time series at a high frequency (12-month moving window, n = 721) into very wet ($\sigma \geq 1$), wet ($0 \leq \sigma < 1$), dry ($-1 \leq \sigma < 0$) and very dry ($\sigma < -1$) periods (Fig. 2a). $\gamma_{CGR}^{T}$ is calculated for each category of periods and we find that it is highest in the very dry period over tropical regions (Fig. 2b). This was previously reported[9],

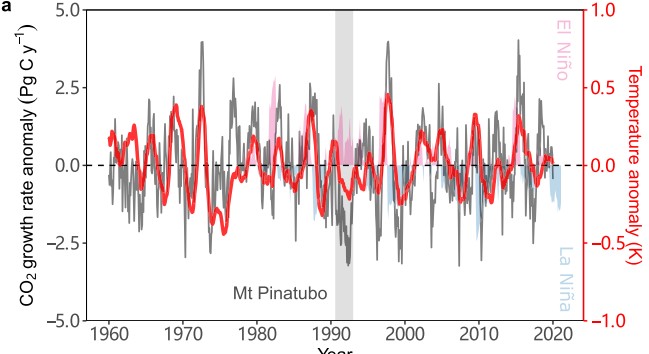

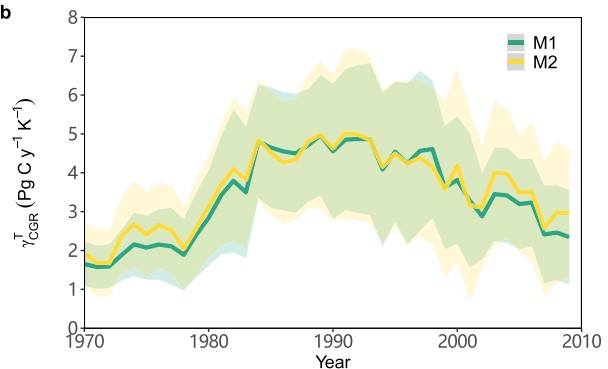

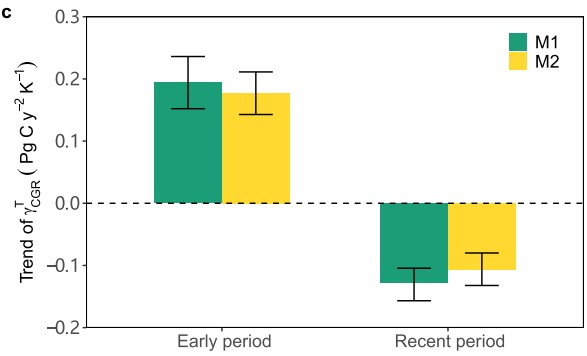

**Fig. 1 | Change in detrended anomalies in $CO_2$ growth rate and tropical temperature. a** Changes in anomalies of detrended $CO_2$ growth rate (CGR) at the Mauna Loa Observatory (black) and in anomalies of detrended tropical temperature (red) derived from the CRU dataset for 1960–2020. Tropical temperature is calculated from the spatial average over vegetated tropical land, 24° N to 24° S. **b** Change in $\gamma_{CGR}^{T}$ for the last six decades. $\gamma_{CGR}^{T}$ is calculated using two multiple regression approaches referring to Eqs. 1–2 (M1 and M2, see "Methods") with the annual CGR and climatic variables in a moving window of 20 years. CGR in the years of 1992–1993 are excluded due to the eruption of Mt Pinatubo. The shaded areas denote 1 SD of the sensitivity derived from a 20-y moving window in 500 bootstrap estimates. The years on the x-axis indicate the central year of the moving window used to derive $\gamma_{CGR}^{T}$. **c** Trends in $\gamma_{CGR}^{T}$ from 1960–1979 to 1979–2000 (early period) and 1980–2001 to 2001–2020 (recent period). Significance ($P < 0.05$) is identified using the Mann-Kendall trend test for both periods. The error bars represent the 95% confidence intervals.

however, the time series of the previous study ended in 2011 and did not include the decreased $\gamma_{CGR}^{T}$ observed in the recent decade.

We further identify that the long-term temporal variations in $\gamma_{CGR}^{T}$ are significantly ($P < 0.05$) correlated with changes in precipitation at a bi-decadal scale (annual precipitation averaged per 20-y moving window) with an average correlation coefficient $r$ of −0.81 (Fig. 2c, d). Partial correlations between $\gamma_{CGR}^{T}$, CGR, temperature and precipitation show that changes in $\gamma_{CGR}^{T}$ are unrelated to changes in temperature (Supplementary Table 1). The significant correlations are identified

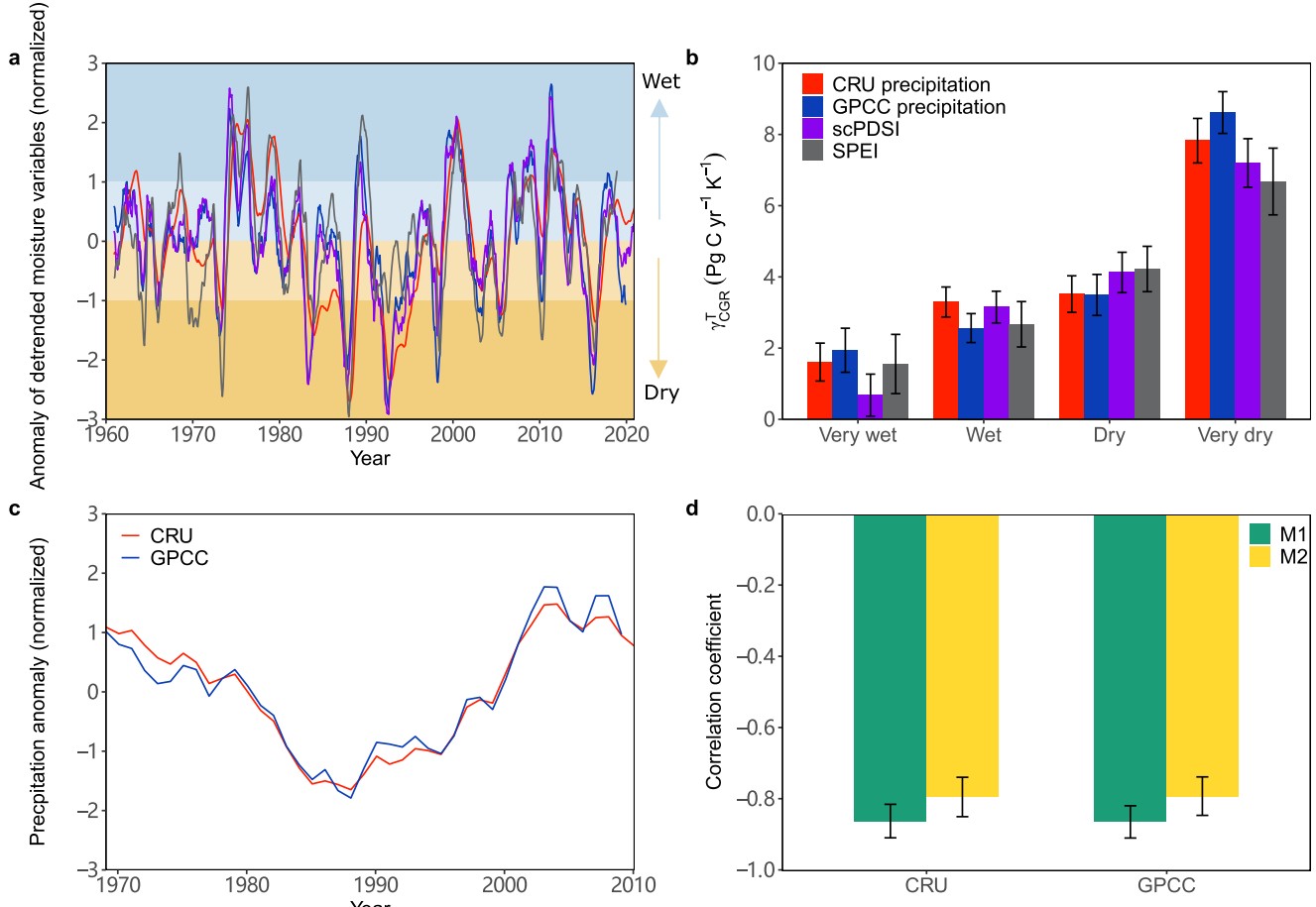

**Fig. 2 | Variations in $\gamma^T_{CGR}$ with varying dryness conditions. a** Variations in long-term anomalies of detrended precipitation, the self-calibrating Palmer Drought Severity Index (scPDSI) and Standardized Precipitation and Evaporation Index (SPEI) calculated from a 12-month moving window ($n = 721$). Water conditions are derived based on the standardized deviation ($\sigma$) of long-term detrended precipitation that are extracted from both CRU and GPCC datasets, scPDSI and SPEI. The long time series are divided into four categories, denoting very wet ($\sigma \geq 1$), wet ($0 \leq \sigma < 1$), dry ($-1 \leq \sigma < 0$) and very dry ($\sigma < -1$) conditions based on the anomalies of detrended precipitation, scPDSI and SPEI, which are shown in different background colors. **b** Average $\gamma^T_{CGR}$ for each category (error bars denote 1 SD of $\gamma^T_{CGR}$ in 500 bootstrap estimates). **c** Variations in the anomalies of bi-decadal precipitation (high frequency precipitation averaged per 20-y moving window). **d** Correlation coefficients between $\gamma^T_{CGR}$ and bi-decadal precipitation. M1 and M2 denoted that $\gamma^T_{CGR}$ are calculated based on Eqs. 1 and 2 (error bars denoted 1 SD of the correlation coefficient in 500 bootstrap estimates).

using a nonparametric random phase test with 1000 Monte-Carlo simulations that is robust to serial autocorrelation. Moreover, we identify the significant correlations after controlling for serial auto-correlations by implementing the Cochrane-Ocrutt procedure (Supplementary Table 2) (see Methods), supporting the robustness of the result. The recent decrease in $\gamma^T_{CGR}$ coincides with increased pre-cipitation at a bi-decadal scale in the tropics (Fig. 2c), where increases in bi-decadal precipitation during 1980–2020 occur over 65 and 73% of tropical land based on the two precipitation datasets, respectively (Supplementary Fig. 8). On the contrary, a sharp increase in $\gamma^T_{CGR}$ is probably associated with a decrease in precipitation during the 2015–2016 extreme El Niño, suggesting that severe water stress in a single year causes the increase in $\gamma^T_{CGR}$ calculated over a window of 20 y (Supplementary Fig. 3c and Fig. 2a). These findings, collectively, sug-gest that changes in water availability in the tropics may be the primary driver of the variations in $\gamma^T_{CGR}$.

Additionally, we detect changes in $\gamma^T_{CGR}$ in the four dryness cate-gories separated by the standardized deviation of the mean annual precipitation, soil moisture, scPDSI, SPEI and terrestrial water storage in the spatial domain (see Supplementary Fig. 9 for the spatial dis-tributions of these long-term average variables). $\gamma^T_{CGR}$ varies greatly between these spatial categories, showing different responses in tro-pical regions with distinct differences in dryness (Supplementary

Figs. 10 and 11). This indicates that regional changes in water avail-ability might contribute to the variations of $\gamma^T_{CGR}$. These climatic and hydrological conditions are changing over time (Supplementary Fig. 12), which is consequently expected to drive temporal variations in $\gamma^T_{CGR}$.

We subsequently use the dynamic ecosystem model LPJ-GUESS to separate, by factorial simulations, the relative importance of changes in each climatic variable (precipitation and solar radiation), atmospheric $CO_2$ concentration, nitrogen deposition and land cover change to the changes in the sensitivity of the carbon cycle to variations in tropical temperature. By running different factorial simulations using a process-based model, it is possible to infer information about the importance of individual driver variables in respect to their respective influence on the dynamics of the carbon cycle[24]. For the full run (including all drivers varying over time, referring to the scenario 1 (SCE1) in Supplementary Table 3) the temporal variations in the sensitivity of modeled net biome productivity (NBP) to tropical temperature ($\gamma^T_{NBP}$) are significantly ($P < 0.05$) correlated with changes in $\gamma^T_{CGR}$, with correlation coefficients of −0.68 for CRU climate data and −0.87 for ERA5 climate data, respectively (Supplementary Figs. 13 and 14). Overall, $\gamma^T_{CGR}$ shows a stronger variation during recent years, and simulated decline in $\gamma^T_{NBP}$ starts earlier than the decline in observed $\gamma^T_{CGR}$ (Fig. 3a). However, the general agreement suggests that the interannual variation of CGR is

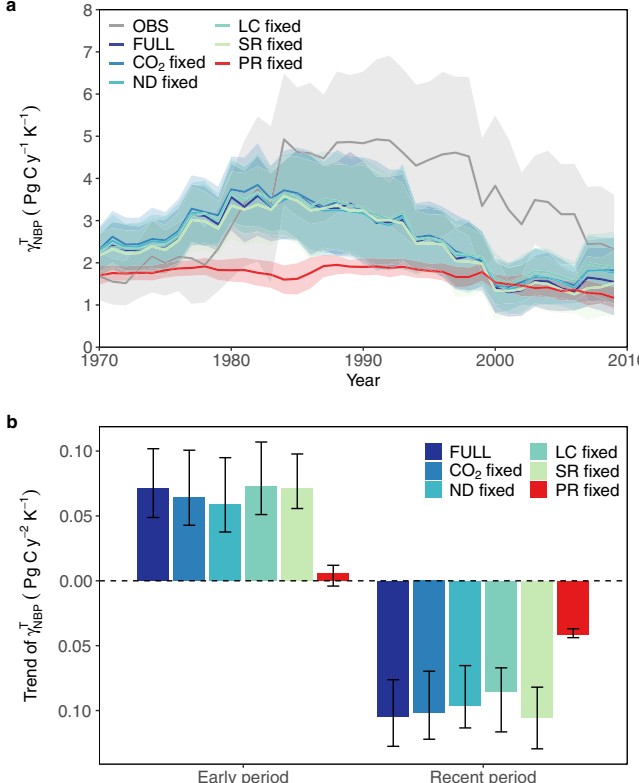

**Fig. 3 | Changes in $\gamma_{NBP}^{T}$ simulated in the LPJ-GUESS model. a** Changes in $\gamma_{CGR}^{T}$ under different simulations for 1960–2020. The details of these simulations is shown in Supplementary Table 3. OBS denotes observed $\gamma_{CGR}^{T}$ based on CGR from the Mauna Loa station. FULL refers to SCE1 and denotes that all driver variables vary with time. $CO_2$ fixed (SCE4): $CO_2$ kept constant and the remaining driver variables vary with time. Same as above, but with nitrogen deposition (ND), land cover change (LC), solar radiation (SR) and precipitation (PR) kept constant. For $CO_2$ and LC annual mean values are used and for ND, SR, PR monthly climatology values (1960–2020) are used. The shaded areas denote 1 SD of the sensitivity derived from a 20-y moving window in 500 bootstrap estimates. The sign of NBP is inverted to be comparable with CGR. **b** Trends in $\gamma_{NBP}^{T}$ calculated from a 20-y moving window for 1960–1979 to 1979–2000 (early period) and 1980–2001 to 2001–2020 (recent period) under different simulations corresponding to Fig. 3a. The error bars represent 95% confidence intervals.

primarily driven by the tropical land carbon flux (NBP)[24,25] and the simulated NBP can thus be used to explore the interannual changes in the $\gamma_{CGR}^{T}$. Furthermore, we find that the LPJ-GUESS simulated changes in $\gamma_{NBP}^{T}$ are in reasonable agreement with the ensemble mean changes from 15 dynamic global vegetation models (DGVMs) from the TRENDY project (Supplementary Fig. 15), supporting the robustness of the LPJ-GUESS simulated dynamics of $\gamma_{NBP}^{T}$. In addition, according to the simulations of LPJ-GUESS, we find that the relative contribution to changes in tropical $\gamma_{NBP}^{T}$ from the tropical regions of the different continents are dominated by tropical Africa, followed by Asia-Australia and South America (Supplementary Fig. 16).

The impact of each individual variable on the variations in $\gamma_{NBP}^{T}$ is analyzed by comparing SCE1 (full simulation) (Supplementary Table 3) with factorial simulations. $\gamma_{NBP}^{T}$ in scenario 2 (SCE2, with precipitation kept constant at its long-term mean seasonal cycle and the remaining drivers varying with time) profoundly differ from the SCE1 and show limited interannual variability and trends (Fig. 3a, b), suggesting that the variation of $\gamma_{NBP}^{T}$ is primarily driven by interannual changes in precipitation. For the remaining runs with one of the variables $CO_2$, nitrogen deposition, land cover or solar radiation being kept constant and the others varying over time, the simulated NBP shows similar variations as the results from SCE1 (Fig. 3).

We further study the sensitivity of changes in $\gamma_{NPP}^{T}$, $\gamma_{Rh}^{T}$, $\gamma_{FIRE}^{T}$ to $\gamma_{NBP}^{T}$ and find that changes in $\gamma_{NBP}^{T}$ is most sensitive to $\gamma_{NPP}^{T}$, followed by $\gamma_{Rh}^{T}$ and $\gamma_{FIRE}^{T}$ (Fig. 4a). We analyze how precipitation regulates changes in the sensitivity of carbon fluxes to tropical temperature using the outputs from LPJ-GUESS. Changes in the sensitivity is simulated under scenarios of SCE1 (see scenario in Supplementary Table 3) where all drivers of carbon fluxes are varying with time and SCE1–SCE2 where precipitation controls the changes, respectively. We reveal a widespread spatial pattern of significant ($P < 0.05$) correlations of $\gamma_{NBP}^{T}$, $\gamma_{NPP}^{T}$, $\gamma_{Rh}^{T}$, $\gamma_{FIRE}^{T}$ between SCE1 and SCE1–SCE2 (Fig. 4b and Supplementary Fig. 17), which support our findings of the importance of precipitation in regulating changes in the sensitivity of carbon fluxes to temperature.

Finally, we explore the variations in $\gamma_{NBP}^{T}$ simulated by the land surface models being part of the ESMs included in CMIP6 (Supplementary Table 4). We calculate the sensitivity of the modeled NBP to tropical temperature and find that most of the ESMs (29 of 33) identify the strong coupling between annual NBP and temperature over tropical land (Supplementary Fig. 18). We further identify 27 ESMs with significant correlations ($P < 0.05$) between tropical NBP and temperature under very wet and very dry conditions determined by the standard deviation ($\sigma$) of the detrended precipitation. The ensemble mean of $\gamma_{NBP}^{T}$ of the 27 ESMs during very dry years ($\sigma < -1$) is higher than that during very wet years ($\sigma \geq 1$) (Fig. 5a) showing agreements with our observed patterns, but the differences in $\gamma_{NBP}^{T}$ between very dry and very wet conditions vary largely among ESMs (Fig. 5b). Of the 27 simulation results that are analyzed, 14 have a higher $\gamma_{NBP}^{T}$ under very dry condition than for wet condition, consistent with the observed $\gamma_{CGR}^{T}$ under very dry condition (Fig. 5b). These results indicate that ESMs can only to some extent capture the response of the carbon flux to changes in temperature under very wet and dry conditions.

## Discussion

We document an increase and decrease in the interannual $\gamma_{CGR}^{T}$ (or land carbon sink), between 1960–2000 and the recent quartet of decades 1980–2020. The increase in the early period has previously been reported[9], whereas the decrease since 1980 documented here needs to be understood in the context of continued warming and increased extreme climate events. The observed decreased $\gamma_{CGR}^{T}$ could indicate a decoupling of the impact of the tropical temperature variations on the carbon cycle. This further suggests that a change in the complex interplay of drivers regulating the exchange of carbon between land and the atmosphere may have taken place[26]. Recent studies have reported that a feedback between soil moisture and the atmosphere primarily controlled the variability and long-term trend of the global terrestrial carbon sink[27,28] and that the role of terrestrial water storage in tropical regions may increase[29]. Moreover, existing studies show that semi-arid ecosystems dominate the interannual variability of global carbon cycle[24,25] and that tropical extreme droughts are the cause for an increase in CGR variability[11]. These studies collectively suggest that water availability plays an important role in regulating the global carbon cycle. Our results show that water availability largely controls the bi-decadal variations in $\gamma_{CGR}^{T}$, showing that the increasing $\gamma_{CGR}^{T}$ in the early years is driven by increased water stress. The decreased $\gamma_{CGR}^{T}$ is considered plausible, as alleviated water stress observed in recent decades could promote photosynthesis that sequesters more atmospheric $CO_2$. These findings suggest that it is the interactions between water and temperature that control the carbon cycle in the tropical regions. Lastly, the simulation results show that water dryness controlling changes in $\gamma_{NBP}^{T}$ are primarily determined by $\gamma_{NPP}^{T}$, followed by $\gamma_{Rh}^{T}$ and $\gamma_{FIRE}^{T}$, suggesting that plant productivity is still key in regulating $\gamma_{CGR}^{T}$. Still, model limitations exist in capturing the dynamics of $\gamma_{CGR}^{T}$ across the tropics, which could thereby lead to uncertainties in the assessment of primary drivers for $\gamma_{CGR}^{T}$.

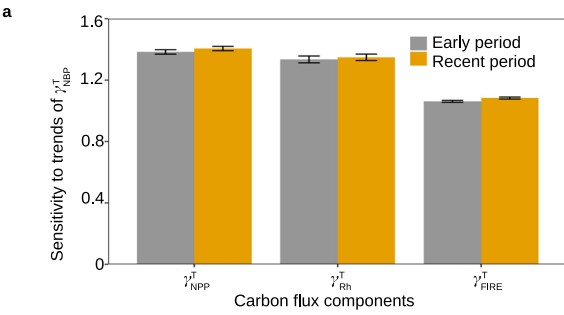

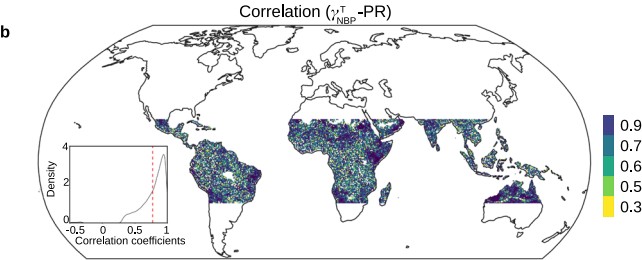

**Fig. 4 | Carbon flux component sensitivity to tropical temperature in relation to precipitation as simulated by LPJ-GUESS. a** Sensitivity of changes in $\gamma^{T}_{NPP}$, $\gamma^{T}_{Rh}$ and $\gamma^{T}_{FIRE}$ to $\gamma^{T}_{NBP}$. The sensitivity is calculated as the regression coefficients for each explanatory variable in a multiple linear regression with the response variable being trends in $\gamma^{T}_{NBP}$ and the explanatory variables being trends in $\gamma^{T}_{NPP}$, $\gamma^{T}_{Rh}$, and $\gamma^{T}_{FIRE}$. The error bars denote standard error of the mean. **b** Spatial patterns of correlation coefficients of $\gamma^{T}_{NBP}$ under scenarios of SCE1 (all driver variables of NBP are varying with time) and SCE1–SCE2 (precipitation drives variations in NBP). The pixels with significant ($P < 0.05$) correlation are shown. Density distribution of correlation coefficients is inserted (red dashed line denotes mean of correlation).

Several other factors, however, may also contribute to changes in $\gamma^{T}_{CGR}$. Possible acclimation of terrestrial ecosystem carbon fluxes to the ongoing warming in tropical regions may contribute to the decreased sensitivity of carbon to temperature[14–16]. Changes in $\gamma^{T}_{CGR}$ could also be associated with the fact that water-driven temporal variations in vegetation productivity and respiration balance locally, leaving a seemingly dominant effect of temperature on the variations in net carbon sink[30]. Moreover, an increase in the frequency of climatic extremes and disturbances such as changes in wildfires and insect outbreaks, and decreasing availability of a necessary nutrient supply (e.g., phosphorus) could influence plant productivity that might indirectly alter the response of the carbon cycle to temperature. We finally show that the majority of ESMs could to only some extent capture the differences in $\gamma^{T}_{NBP}$ under very dry and very wet conditions, and a better constrain of the spread of the models on carbon cycle sensitivity to temperature is still needed to reduce the uncertainty in the predictions of the future terrestrial carbon uptake[5]. Improved knowledge on the complex interplay of processes regulating the response of the terrestrial carbon cycle to drought will require further research.

Changes in climate have altered terrestrial ecosystems in recent decades. In particular, the changes in temperature and precipitation seem to a large extend to regulate the exchange of carbon between land and the atmosphere, sustaining the important role of terrestrial ecosystems in the climate system. Our results provide clear evidence that water availability exerts an important control on the observed bi-decadal variations in CGR sensitivity to tropical temperature. This finding highlights the importance of the water-temperature interactions in regulating the CGR over the tropics and provides insights to our understanding of climate-carbon interactions.

## Methods

### Atmospheric $CO_2$

The monthly mean atmospheric carbon dioxide concentration ($CO_2$) at the Mauna Loa Observatory for 1959–2020 and the South Pole station for 1980–2020 were accessed from the US National Oceanic and Atmospheric Administration (http://www.esrl.noaa.gov/gmd/ccgg/trends/). The Mauna Loa Observatory $CO_2$ dataset represents the longest record of direct measurements of $CO_2$ in the atmosphere. There are many $CO_2$ observations missing in the South Pole station dataset before 1980, and we have thus chosen the time series of $CO_2$ observations from 1980 to 2020, which allowed us to detect the changes CGR sensitivity for the recent period.

### Climatic data

We extracted monthly temperature, precipitation and solar radiation (cloud cover in CRU-4) for 1960–2020 from both the CRU-4 and ERA5 datasets. Monthly climatic data at a spatial resolution of 0.5° from the Climate Research Unit (CRU TS 4.05), University of East Anglia, are generated by scaling up rain-gauge observations[31]. We resampled the data to a spatial resolution of 0.25° using bilinear interpolation. The ERA5 data were downloaded from the website of the European Centre for Medium-Range Weather Forecasts (ECMWF) Reanalysis (ERA5)[32], which are the fifth generation ECMWF reanalysis data for the predictions of global climate and weather replacing the ERA-Interim reanalysis. The data are provided at a spatial resolution of 0.1° and a temporal coverage from 1950 to the present. We also extracted monthly gauges-based precipitation data for 1960–2019 from the Global Precipitation Climatology Centre (GPCC)[33]. We further used the self-calibrating Palmer Drought Severity Index (scPDSI) for the detection of changes in water stress conditions[34], where negative values of scPDSI indicate drier conditions and vice versa. scPDSI is estimated based on precipitation, potential evapotranspiration (PET) and temperature, and the Penman-Monteith method is used to calculate PET. Likewise, the Standardized Precipitation and Evaporation Index (SPEI) data (using a three-month integration period) for 1960–2018 were extracted to indicate water availability conditions[35]. We used monthly root-zone soil-moisture data from the dataset of the Global Land Evaporation Amsterdam Model version 3 (GLEAM-v3) produced at a spatial resolution of 0.25° and with a temporal coverage of 1980–2020[36]. We also extracted reconstructed data for terrestrial water storage driven by ERA5 climatic data[37], which included GSFC-ERA5 and JPL-ERA5 with a temporal coverage of 1979–2018. It should be noted that spurious trends might occur in these reanalysis data due to the discontinuity of the assimilated observations[9].

### LPJ-GUESS model

LPJ-GUESS[38] is a dynamic vegetation model that simulates the cycling of carbon and nitrogen within vegetation and soil. Vegetation is represented with 12 plant functional types[39], coexisting in patches of vegetation that undergo stochastically varying disturbances. Net biome productivity (NBP) was obtained from a simulation forced with monthly gridded data at a spatial resolution of 0.5 × 0.5° from CRU TS 4.05[31], monthly model-derived estimates of nitrogen deposition and annual atmospheric $CO_2$ concentration based on ice-core data and atmospheric observations in a simulation for 1901–2020. Land use was represented applying historical reconstructions of land use from previously published data[39]. In addition to this simulation, a set of factorial simulations (Supplementary Table 3) was performed, in which individual drivers (temperature, precipitation, cloudiness, atmospheric $CO_2$ concentration, N deposition or land use) were kept constant at the mean conditions for 1960–2020, while all other drivers were time-variant as in the main simulation. We subsequently extracted time series of carbon flux components namely net primary production (NPP), heterotrophic respiration ($R_h$), and fire emission (FIRE) from these simulations.

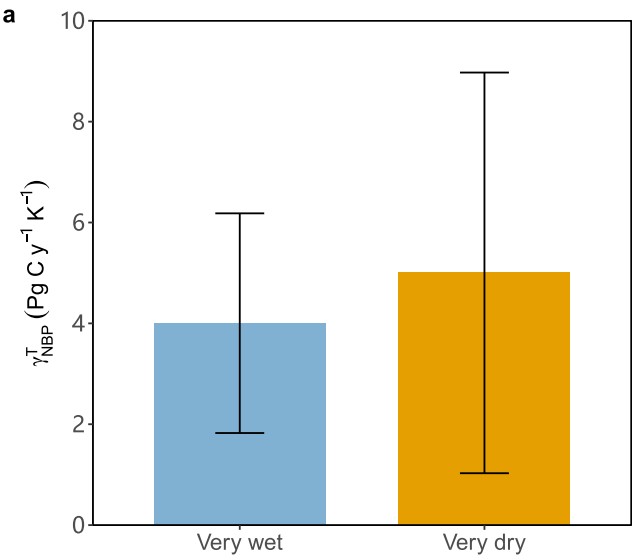

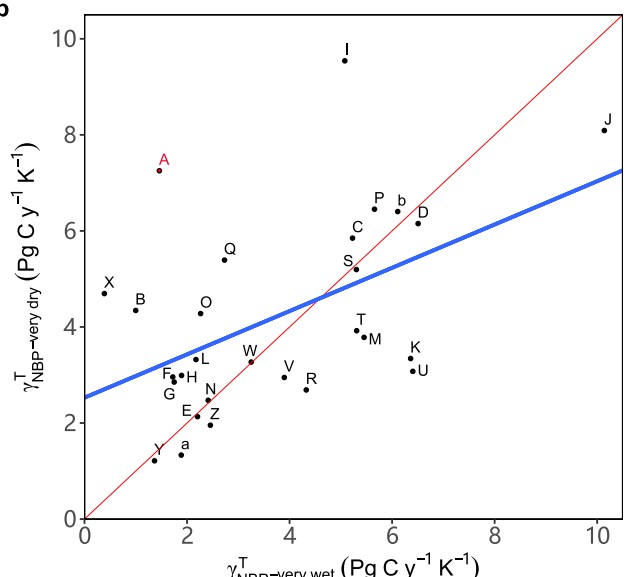

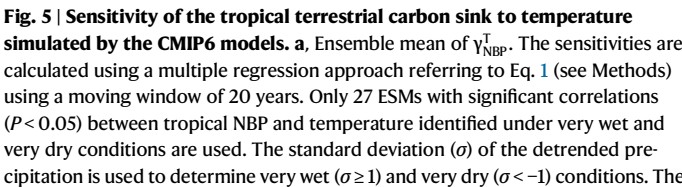

**Fig. 5 | Sensitivity of the tropical terrestrial carbon sink to temperature simulated by the CMIP6 models. a**, Ensemble mean of $\gamma_{NBP}^{T}$. The sensitivities are calculated using a multiple regression approach referring to Eq. 1 (see Methods) using a moving window of 20 years. Only 27 ESMs with significant correlations ($P < 0.05$) between tropical NBP and temperature identified under very wet and very dry conditions are used. The standard deviation ($\sigma$) of the detrended precipitation is used to determine very wet ($\sigma \geq 1$) and very dry ($\sigma < -1$) conditions. The

sign of NBP is inverted to be comparable with CGR. The error bars denote 1 SD of $\gamma_{NBP}^{T}$ among 27 ESMs. **b**, $\gamma_{NBP}^{T}$ at very dry conditions as a function of $\gamma_{NBP}^{T}$ at very wet conditions. The red line is the 1:1 line and the blue line is the fitted regression line. The point labeled with A in red color denotes observed $\gamma_{CGR}^{T}$ under very wet and dry conditions and the remaining labels of B, C, D... correspond to individual ESMs (shown in Supplementary Table 4).

## CMIP6 outputs

We collected monthly simulated NBP, temperature, precipitation and solar radiation for 1960–2014 from the historical simulations from 33 ESMs (Supplementary Table 4) performed as contribution to CMIP6[40]. Solar radiation was calculated as the difference between surface down- and upwelling shortwave radiation simulated in CMIP6. All CMIP6 outputs were resampled to a spatial resolution of 1°using bilinear interpolation.

## Other data

We extracted monthly NBP data under scenario 3 for 1960–2019 from 15 dynamic global vegetation models (DGVMs) from TRENDY version-9[41] being part of the Global Carbon Project[42]. DGVMs used in this study included CLASSIC, CLM5.0, DLEM, IBIS, ISBA-CTRIP, JSBACH, JULE-ES, LPJ-GUESS, LPX-Bern, OCN, ORCHIDEE-CNP, ORCHIDEEv3, SDGVM, VISIT and YIBs. We further collected data for ESA CCI land cover to exclude water bodies and built-up land in the tropical regions. We also extracted data from Multivariate ENSO Index Version 2 (MEI.v2) to indicate El Niño and La Niña during 1979–2021.

## Analyses

We used the first-order difference of $CO_2$ between two successive months to calculate the monthly $CO_2$ growth rate (CGR). We calculated annual sum of CGR for 1960–2020 ($n = 59$ after removing years of 1992–1993 due to the eruption of Mt Pinatubo) to identify the inter-annual variability and trends in the sensitivity of CGR to tropical temperature ($\gamma_{CGR}^{T}$). This is because the Mt Pinatubo eruption was suggested to exert strong effects on the carbon cycle by increasing diffuse light that promotes photosynthesis[22], as well as causing a decrease in net shortwave radiation that may constrain photosynthesis and causing a reduction in precipitation[43] and cooling[44] that both can affect photosynthesis and heterotrophic respiration. We also calculated CGR at a high temporal frequency using a 12-month moving window ($n = 721$), which facilitates identification of the anomaly of changes in $\gamma_{CGR}^{T}$ related to climate extreme events and increase the

degrees of freedom. CGR was converted from ppm y⁻¹ into PgC y⁻¹ by multiplying with the conversion factor of 2.124 PgC ppm⁻¹[18]. Climate variables of tropical temperature, precipitation and solar radiation over the surface area of vegetated land between 24° N and 24° S based on a land cover map of ESA CCI 2015 were derived at annual and high frequency scales to be consistent with CGR.

$\gamma_{CGR}^{T}$ was defined as the slope of the regression in a multiple regression accounting for the covarying climatic variables of precipitation and solar radiation. Here $\gamma_{CGR}^{T}$ was calculated using a multiple linear regression approach (Eq. 1; further referred to as M1) and a non-linear regression approach accounting for the interaction effects of temperature and precipitation on CGR (Eq. 2; further referred to as M2). All variables were detrended, resulting in anomalies of temperature ($\Delta T$), precipitation ($\Delta P$) and incoming shortwave radiation ($\Delta R$) relative to their long-term linear trend. These anomalies were used to calculate the slope for a 20-year moving window between 1960–2020.

$$CGR = \gamma_{CGR}^{T}\Delta T + \tau_{CGR}^{P}\Delta P + \delta_{CGR}^{R}\Delta R + \varepsilon \tag{1}$$

$$CGR = \gamma_{CGR}^{T}\Delta T + \tau_{CGR}^{P}\Delta P + \delta_{CGR}^{R}\Delta R + \theta_{CGR}^{I}(\Delta T \times \Delta P) + \varepsilon \tag{2}$$

Where $\gamma_{CGR}^{T}$, $\tau_{CGR}^{P}$, $\delta_{CGR}^{R}$ and $\theta_{CGR}^{I}$ are the sensitivities of CGR to temperature, precipitation, incoming radiation and the interactions between temperature and precipitation, and $\varepsilon$ is the residual error.

We calculated $\gamma_{CGR}^{T}$ using a moving window of 20 y during the study period. $\gamma_{CGR}^{T}$ was calculated at annual and high temporal frequency scales. The CGR calculated with shorter windows, such as 15 y, could be affected by autocorrelation and was not considered in our study. A previous study also reported that the different lengths of selected moving windows had little effect on the results[3]. The sensitivities were calculated in a 500-member bootstrap to estimate the uncertainties for each window. Similarity, the sensitivities of NBP ($\gamma_{NBP}^{T}$), NPP ($\gamma_{NPP}^{T}$), $R_h$ ($\gamma_{Rh}^{T}$) and fire emissions ($\gamma_{FIRE}^{T}$) to tropical temperature were calculated.

We explored the variations of $\gamma_{CGR}^T$ under spatio-temporal gradients of water conditions. We first grouped the long-term time series at a high temporal frequency ($n = 721$) of data based on the standard deviation ($\sigma$) of detrended precipitation from CRU and GPCC into four bins. We chose the data at a high frequency in order to increase the degrees of freedom to test for statistical significance of the sensitivity analysis in each bin. These bins were defined as the periods of very wet ($\sigma \geq 1$), wet ($0 \leq \sigma < 1$), dry ($-1 \leq \sigma < 0$) and very dry ($\sigma < -1$) following the ref. [9]. We thus calculated $\gamma_{CGR}^T$ within each bin and a 500-member bootstrap was used to estimate uncertainties for each bin. $\gamma_{CGR}^T$ was also calculated based on the bins divided by detrended anomalies of scPDSI and SPEI. Similarly, we calculated $\gamma_{NBP}^T$ under very wet ($\sigma \geq 1$) and very dry ($\sigma < -1$) conditions with the outputs (i.e., NBP, temperature, precipitation, solar radiation) from ESMs. The very wet and dry conditions were determined by the anomalies of detrended precipitation for 1960–2014 derived from ESMs.

We then calculated $\gamma_{CGR}^T$ based on the spatial groups, where the tropical regions (pixels) were divided into four groups based on the standard deviation ($\sigma$) of mean annual precipitation, soil moisture, scPDSI, SPEI and terrestrial water storage at the pixel level. These groups were defined as the regions of very wet ($\sigma \geq 1$), wet ($0 \leq \sigma < 1$), dry ($-1 \leq \sigma < 0$) and very dry ($\sigma < -1$) to indicate four distinct gradients of water conditions. These variables at the pixel level in each group were aggregated to derive time series of climatic variables for 1978–2020 that were used to calculate $\gamma_{CGR}^T$ based on Eq. 1. The long-term GLEAM soil moisture and reconstructed terrestrial water storage were not used to define the periods with various water stress, as these data were mainly driven by reanalysis data that may have spurious trends[9].

We next designed several simulations to explore the underlying mechanism of changes in $\gamma_{NBP}^T$ using the LPJ-GUESS model. We first ran the simulation with all the driver variables varying with time, defined as scenario 1 (SCE1) (Supplementary Table 3). To attribute a driver variable to the variation in NBP sensitivity, we ran the simulations with precipitation kept constant (SCE2), and so forth for solar radiation (SCE3), $CO_2$ (SCE4), land use (SCE5) and nitrogen deposition (SCE6), with the remaining driver variables time-varying. The driver variables kept constant were assigned the value of their long-term annual mean over 1960–2020 in each scenario. NBP derived from each simulation was used to calculate $\gamma_{NBP}^T$. The use of constant values for driver variables in SCE2-6 potentially leading to large differences in $\gamma_{NBP}^T$ between SCE1 and SCE2-6 was used a means to assess the importance of the individual drivers of changes in $\gamma_{NBP}^T$. Additionally, the relative contributions of the tropical regions of different continents (Africa (AF), Asia-Australia (AA) and south America (SA)) to changes in $\gamma_{NBP}^T$ for the entire tropical region were assessed using a multiple linear regression model. Here $\gamma_{NBP}^T$ of the entire tropical region was set as response variable and $\gamma_{NBP}^T$ of the tropical regions of different continents as explanatory variables. The relative importance is assessed using the "lmg" approach, which is calculated based on a sequential $R^2$ but considers the dependence on ordering of explanatory variables by averaging over the total number of combinations of ordering in a multiple regression.

The Theil–Sen slope and Mann–Kendall trend test were used to identify trends and their significance ($P < 0.05$) in $\gamma_{CGR}^T$ or $\gamma_{NBP}^T$ for the early and recent periods. We used the function 'zyp.trend.vector' provided in the R package 'zyp', including the Yue–Pilon pre-whitening method applied to remove serial autocorrelation[23], to conduct the trend test. The slope of a linear regression between trends in $\gamma_{NBP}^T$ and each of $\gamma_{NPP}^T$, $\gamma_{Rh}^T$, and $\gamma_{FIRE}^T$ (carbon flux components) were used to detect the sensitivity of changes between variables. We used partial Spearman's rank correlation analysis to control for the covarying effects and identified the coupling strength between variables. We also used Spearman's rank correlation method to measure the coupling strength between different variables. Spearman's rank correlation is a nonparametric measure

of correlation between two variables and the significance of correlations was estimated here using a nonparametric random phase test with 1000 Monte-Carlo simulations, which are robust to serial autocorrelation[45]. Moreover, we used the Cochrane-Ocrutt procedure to remove the impacts of serial autocorrelation on the relationships established between these time series variables (Supplementary Table 2). The response and explanatory variables were adjusted by subtracting the previous value multiplied by $\rho$ (the first-order autocorrelation parameter) from the original values and these adjusted values were used in a multiple linear regression, with $\gamma_{CGR}^T$ as response variable and CGR, PR, TMP as explanatory variables. For example, PR was adjusted as RR_adjusted = $PR_i - \rho PR_{i-1}$. The Durbin-Watson indicator (DWI) was used to detect the autocorrelation at lag 1 in the residuals ($e_i$) from the multiple linear regression of a target variable against time[46] using a significance level of $P < 0.001$. DWI was calculated through the Eq. (3) as follows:

$$\text{DWI} = \frac{\sum_{i=2}^{n} (e_i - e_{i-1})^2}{\sum_{t=1}^{n} e_i^2} \tag{3}$$

Where $n$ is the number of observations, $e_i$ is the $i$th residual of the linear regression of the investigated variable against time. The DWI values range between 0 and 4. A value of 2 indicates no autocorrelation, whereas values from 0 to 2 indicate positive autocorrelation and values from 2 to 4 denote negative autocorrelation in the time series of observations.

## Data availability
All data used to support the findings of this study are publicly available. Atmospheric $CO_2$ concentration at the Mauna Loa Observatory and the South Pole station are available from http://www.esrl.noaa.gov/gmd/ccgg/trends/. CRU climatic data are available from https://crudata.uea.ac.uk/cru/data/hrg/. GPCC precipitation are available from https://www.dwd.de/EN/ourservices/gpcc/gpcc.html ERA5 climatic data are available from https://cds.climate.copernicus.eu/cdsapp#!/dataset/reanalysis-era5-land-monthly-means?tab=overview. CMIP6 outputs are available from https://esgf-node.llnl.gov/search/cmip6/. scPDSI and SPEI data are available from https://crudata.uea.ac.uk/cru/data/drought/ and https://spei.csic.es/database.html. GRACE data are available from https://figshare.com/articles/dataset/GRACE-REC_A_reconstruction_of_climate-driven_water_storage_changes_over_the_last_century/7670849. LPJ-GUESS model simulations are available from Guy Schurgers (gusc@ign.ku.dk) upon request. The outputs of DGVMs from TRENDY are available from Hui Yang (huiyang.pku@gmail.com) upon request.

## Code availability
Python code for processing the data and R code for generating the figures are available from the corresponding author upon request.

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

## Acknowledgements

W.M.Z. acknowledged funding from the National Natural Science Foundation of China (grant number: 42001349). W.M.Z. and M.B. were supported by ERC project TOFDRY (grant number: 947757).

## Author contributions

W.M.Z. conceived the study, conducted data analysis and wrote the first draft of manuscript. G.S., J.P., M.B., X.T., P.C. and R.F. aided in the discussion of the results. G.S. and J.T. aided in the simulations of LPJ-GUEE under different scenarios. H.Y. contributed to the DGVMs outputs from TRENDY. W.M.Z., G.S., J.P., R.F., H.Y., J.T., X.T., P.C. and M.B. contributed to the interpretation of the results and to the text.

## Competing interests

The authors declare no competing interests.
