## [Peer Review File · Nature Communications]

nature portfolio

Peer Review File

Recent decrease of the impact of tropical temperature on the carbon cycle linked to increased precipitationREVIEWER COMMENTS

Reviewer #1 (Remarks to the Author):

This paper addresses an interesting and important question regarding the “changing” sensitivity of global atmospheric CO₂ growth rate (CGR) to variations of tropical land temperature. It reports that the CGR-temperature sensitivity increased by ~200% between 1960-2000 (which was reported by some previous studies) but then decreased back to its 1960 level by 2020. The paper argues that such variations of the CGR-temperature variability are mainly regulated by corresponding variations in tropical precipitation.

I think this study is onto some important results BUT have concerns about their current arguments. A 200% variation of CGR-temperature sensitivity at the global scale in a few decades is HUGE! I am not fully convinced that the variations of tropical precipitation alone are strong enough to drive such changes. The modeling results presented by the paper didn't elucidate the underlying biogeophysical mechanisms, either. My specific comments/questions are as follows:

1) The statistical models used in this study, as suggested by Eq. 1 or Eq. 2, are not complete but neglect some important terms. From the point of global carbon cycling it is quite straightforward to see that

$$C \text{ Sink} = \text{Emission} - \text{CGR} = \text{Beta} * A_{\text{co2}} - \text{Gamma} * T + O(\text{errors})$$

where Beta is the CO₂ fertilization effect, A_{co2} is the atmospheric CO₂ concentration, and Gamma (or G) is the CGR sensitivity to T. We can decompose the variables to the climatology (~20yr moving mean) and the interannual variations as

$$(E_{\text{clt}} + E_{\text{iav}}) - (\text{CGR}_{\text{clt}} + \text{CGR}_{\text{iav}}) = \text{Beta} * (A_{\text{clt}} + A_{\text{iav}}) - (G_{\text{clt}} + G_{\text{iav}}) * (T_{\text{clt}} + T_{\text{iav}}) + O(\text{errors})$$

Note that because Gamma (G) is not a constant, it has to be decomposed to climatological and interannual components as temperature. Subtracting climatologies terms and neglecting smaller variables (e.g., E_{iav} and A_{iav}) leads to

$$\text{CGR}_{\text{iav}} = G_{\text{clt}} * T_{\text{iav}} + G_{\text{iav}} * T_{\text{iav}} + G_{\text{iav}} * T_{\text{clt}} + O(\text{errors}) = G * T_{\text{iav}} + G_{\text{iav}} * T_{\text{clt}} + O(\text{errors})$$

So we see the statistical models of Eq. 1 or 2 missed, at least, the G_{iav}*T_{clt} term. The incompleteness of the model could contribute to the over-/under-estimation of G_{iav}.

2) Biogeophysical/chemical interpretation: In the LPJ-GUESS model experiments, what carbon flux component mainly contributes to Gamma*T? Is it respiration (vegetation or/and soil carbon) or photosynthesis (GPP)? How exactly precipitation variations regulate these processes? Using only NBP cannot answer these questions.

As such, I cannot recommend the publication of the paper on Nature Communications in its current form.

Reviewer #2 (Remarks to the Author):

This paper focused on the recent decrease in the sensitivity of the atmospheric CO₂ growth rate (CGR), and the authors combine data analysis and process-based model to show that the weaker sensitivity of CGR is controlled by increases in precipitation. CGR sensitivity has been extensively studied in the literature, including multiple high-profile papers. The finding regarding the recent decline in CGR sensitivity is new, and the authors did a good job analyzing and explaining the causes. Analyses methods used in this paper are mostly proven techniques reported in Wang et al. (2014) and Luo et al. (2022), and I have no issues regarding the robustness of the results. However, linkage between increased precipitation and decrease in CGR sensitivity could already be expected from previous research, so it would be better if new knowledge gained from

analyses here could be better highlighted. While simulation results from process-based models were reported, it would make the paper stronger if they were analyzed further to advance our mechanistic understandings of the reported phenomenon.

One area I would like to see improved is process understanding based on the simulation results. From the simulation results, LPJ-GUESS shows a clear signal of precipitation-driven changes in CGR sensitivity, but results from CMIP6 seem quite mixed. Is the mean sensitivity significantly different between the very wet and very dry conditions in Fig. 4a? In Fig. 4b, we see similar number of models above/below the 1:1 line. What might be the reasons for some CMIP6 models to exhibit quite different behavior compared to LPJ-GUESS model? Several factors such as the acclimation of terrestrial ecosystem to ongoing warming and increase in disturbances were mentioned, but since the acclimation effect is not represented in process-based models (is this right?), it is difficult to quantify its contribution to changes in CGR sensitivity. However, could disturbance be held fixed in one of the factorial experiments to analyze its role?

Secondly, while the paper in general is easy to follow, my impression is that the methods were not explained as clearly as in some other cited papers. For example, Luo et al. (2022) explained the Durbin-Watson indicator in detail, whereas this paper only mentioned the method was used. For readers familiar with the literature it is not an issue, as no new methodology was introduced. Other readers however, would have to look into multiple cited paper sources to understand fully some of the methodology. It would improve the readability if various techniques were better explained in this paper, similar to other papers.

Lastly, unlike Luo et al. (2022), changes in water sensitivity of CGR or NBP were not reported in this paper. I wonder if additional insights could be gained from such analyses on the observed and simulated data.

Some minor grammatical issues:

P1L37: "most challenging tasks" to simulate correctly?

P8L218: "reversal"-> "a reversal"

P9L252: "evidence" -> "provide evidence"

Reviewer #3 (Remarks to the Author):

Review of “Recent decrease of the impact of tropical temperature on the carbon cycle linked to increased precipitation”, by Zhang et al., submitted to Nature Communications (Manuscript #370727)

This is an interesting paper that diagnoses changes in the sensitivity of the growth-rate of atmospheric CO₂ to tropical temperature variations (γ_{CGR}), and presents evidence that these are due to changing moisture availability in the tropics. It follows on from previous *Nature* papers which (a) identified that this metric is well correlated with projected carbon loss in the tropics due to future warming (Cox et al., 2013); and (b) showed evidence that γ_{CGR} had increased since 1960 (Wang et al., 2014).

This Zhang et al. submission suggests that γ_{CGR} (as calculated within 20 year windows) has decreased since 1990 and is now back close to its 1970 level. The reason for this decline is traced to a minimum in tropical precipitation around 1990. The paper is relevant to climate change and climate variability and its impact on tropical land carbon stores. It is therefore potentially of broad general interest to the readership of *Nature Communications*.

However, I think the current draft needs to be improved in a number of ways before it should be published:

- 1) **Mechanisms:** the paper currently offers little explanation for the changes in γ_{CGR} aside from the correlation with overall tropical mean precipitation (and the related PDSI). What are the possible mechanisms that would lead to a dependency of the sensitivity of γ_{CGR} ? How could you distinguish between these possible mechanisms? Wouldn't you expect γ_{CGR} to be roughly proportional to GPP? If so, shouldn't γ_{CGR} increase (rather than decrease) with less drought-limited GPP? How does your suggestion of reducing moisture stress in the tropics compare to other recent papers suggesting drying and reducing resilience of the Amazon rainforest (e.g. Boulton et al., 2022)? *These issues need to be discussed somewhere in a revised paper – ideally in the Discussion section which is currently a little repetitive;*
- 2) **Spatial Detail:** one way to separate out the mechanisms is to look not just at means for the whole tropics, but instead to look also at the spatiotemporal variability of the climate and how this is correlated with the spatiotemporal variability of net carbon uptake. This is obviously difficult from observations alone (because we do not have measurements of carbon uptake for each gridpoint in the real world), but it can be done with models. I notice also that there are maps in the Supplementary Material. *Some explanatory spatial detail (maps etc.) needs to be included in the main manuscript;*
- 3) **DGVM Modelling results:** it is good to see an attempt at corroboration with model results, although this is currently inconclusive. A single DGVM model (LPJ-GUESS) is used, but we know that models differ in their responses to climate variations (with no obvious best model). *I advise the authors to instead use data from each of the Trendy models;*

- 4) **CMIP6 Model results:** the CMIP6 models shown in Figure 4 show no significant difference between γ_{CGR} for 'very wet' and 'very dry' years (note overlapping uncertainty bars in Figure 4a), although the manuscript underplays this ("These results indicated that ESMs can to some extent capture the response of the carbon flux to changes in temperature under very wet and dry conditions"). *The comparison between models and observations needs to be much clearer, e.g. by including equivalent observational estimates on Figure 4(a) and 4(b);*
- 5) **Clarity of model data comparisons:** in general the model data comparisons are not made very clearly. *I advise the authors to include their observational estimates on Figures 3(a) and 4, and to include a best fit straightline on Figure 4(b) in addition to the one-to-one line. The points in Figure 4(b) are also very difficult to see because of the long model names on the plot. Please remove these model names and instead include distinguishable symbols (e.g. 'a', 'b', 'c' etc,) and a key;*
- 6) **Clarity of the language:** in general the paper is well written but there are some typos that need correcting. Given the other more significant revisions listed above, I don't intend to go through these one by one now, but will do so if I am asked to review a revised manuscript.

Overall, I think this is an interesting study, and is of appropriately broad interest for *Nature Comm.*, but needs these fairly major revisions before publication.

REVIEWER COMMENTS

Reviewer #1 (Remarks to the Author):

This paper addresses an interesting and important question regarding the “changing” sensitivity of global atmospheric CO₂ growth rate (CGR) to variations of tropical land temperature. It reports that the CGR-temperature sensitivity increased by ~200% between 1960-2000 (which was reported by some previous studies) but then decreased back to its 1960 level by 2020. The paper argues that such variations of the CGR-temperature variability are mainly regulated by corresponding variations in tropical precipitation.

I think this study is onto some important results BUT have concerns about their current arguments. A 200% variation of CGR-temperature sensitivity at the global scale in a few decades is HUGE! I am not fully convinced that the variations of tropical precipitation alone are strong enough to drive such changes. The modeling results presented by the paper didn't elucidate the underlying biogeophysical mechanisms, either.

Response: We would like to thank reviewer for the positive evaluation of the importance of the study and for the constructive comments. Following these, we have now made some important revisions to the manuscript, which we believe have improved the work.

The sensitivity of the CGR to tropical temperature by approximately 200% from 1960–1979 to 1979–2000 has already been reported in two previous studies (ref 1-2).

1.Wang, X., Piao, S., Ciais, P., Friedlingstein, P., Myneni, R.B., Cox, P., Heimann, M., Miller, J., Peng, S., Wang, T., Yang, H., Chen, A., 2014. A two-fold increase of carbon cycle sensitivity to tropical temperature variations. *Nature* 506, 212–215.

2.Luo, X., Keenan, T.F., 2022. Tropical extreme droughts drive long-term increase in atmospheric CO₂ growth rate variability. *Nat. Commun.* 13, 1193.

Our results corroborate these findings (including the similar magnitudes of the changes– see also the supporting comment from reviewer #2 on this particular point) serving as the point of departure for our new study.

In this revision, we have recalculated the sensitivity by accounting for ~20 yr moving mean of climate variables and CGR (namely considering the sensitivity of CGR to temperature climatology), which however result in insignificant differences in the CGR sensitivity (see the following response below) compared with the results without using climatology data.

To elucidate the underlying mechanisms, we have now further compared the temperature sensitivities of NPP, soil respiration and fire emissions between the standard run of LPJ-GUESS (full run) and the run with no precipitation interannual variability, and found a dominant control of precipitation on the sensitivities of these three carbon flux components to temperature. Spatial distributions of the correlations between precipitation and these carbon components also support the dominant role of precipitation controlling the variations of CGR sensitivity. Moreover, previous studies have also suggested the important role of wet conditions/soil moisture (largely related to precipitation in our study) in controlling the variation in carbon fluxes (ref 3-5).

3. Green, J.K., Seneviratne, S.I., Berg, A.M., Findell, K.L., Hagemann, S., Lawrence, D.M., Gentine, P., 2019. Large influence of soil moisture on long-term terrestrial carbon uptake. *Nature* 565, 476–479.
4. Ahlström, A., Raupach, M.R., Schurgers, G., Smith, B., Arneth, A., Jung, M., Reichstein, M., Canadell, J.G., Friedlingstein, P., Jain, A.K., Kato, E., Poulter, B., Sitch, S., Stocker, B.D., Viovy, N., Wang, Y.P., Wiltshire, A., Zaehle, S., Zeng, N., 2015. The dominant role of semi-arid ecosystems in the trend and variability of the land CO₂ sink. *Science* (80-.). 348, 895–899.
5. Humphrey, V., Berg, A., Ciais, P., Gentine, P., Jung, M., Reichstein, M., Seneviratne, S.I., Frankenberg, C., 2021. Soil moisture – atmosphere feedback dominates land carbon uptake variability. *Nature* 592.

We have now included an analysis related to the biogeophysical explanation on changes in CGR sensitivity using the outputs from LPJ-GUESS model in the revised manuscript. Please also see the details below in response to comment 2.

My specific comments/questions are as follows:

1) The statistical models used in this study, as suggested by Eq. 1 or Eq. 2, are not complete but neglect some important terms. From the point of global carbon cycling it is quite straightforward to see that

$$C \text{ Sink} = \text{Emission} - \text{CGR} = \text{Beta} * A_{\text{co2}} - \text{Gamma} * T + O(\text{errors})$$

where Beta is the CO₂ fertilization effect, A_{co2} is the atmospheric CO₂ concentration, and Gamma (or G) is the CGR sensitivity to T. We can decompose the variables to the climatology (~20yr moving mean) and the interannual variations as

$$(E_{\text{clt}} + E_{\text{iav}}) - (\text{CGR}_{\text{clt}} + \text{CGR}_{\text{iav}}) = \text{Beta} * (A_{\text{clt}} + A_{\text{iav}}) - (G_{\text{clt}} + G_{\text{iav}}) * (T_{\text{clt}} + T_{\text{iav}}) + O(\text{errors})$$

Note that because Gamma (G) is not a constant, it has to be decomposed to climatological and interannual components as temperature. Subtracting climatologies terms and neglecting smaller variables (e.g., E_{iav} and A_{iav}) leads to

$$\text{CGR}_{\text{iav}} = G_{\text{clt}} * T_{\text{iav}} + G_{\text{iav}} * T_{\text{iav}} + G_{\text{iav}} * T_{\text{clt}} + O(\text{errors}) = G * T_{\text{iav}} + G_{\text{iav}} * T_{\text{clt}} + O(\text{errors})$$

So we see the statistical models of Eq. 1 or 2 missed, at least, the G_{iav}*T_{clt} term. The incompleteness of the model could contribute to the over-/under-estimation of G_{iav}.

Response: Thanks for this comment. For the response, we have decomposed these variables (climate and CGR) into interannual variations and climatology (20 year moving mean) to calculate CGR sensitivities in response to short-term (annual) and long-term (20-year) changes. We found no statistically significant difference (p = 0.8962) in Gamma_{iav} (γ_{CGR}^T , calculated sensitivity) calculated by our initial approach and the approach proposed by the reviewer, based on the two-sided Student's t-test and the result is shown below in the Fig. R1. (R denotes figures presented in the response letter) (M1 denote the Gamma_{iav} calculated using Eq 1 in our manuscript and the line called “Account for climatology” show the result of the calculation after accounting for climatology). To be consistent with previous publications on the subject, we therefore propose not to separate climatology + interannual variations when calculating Gamma_{iav} (due to its insignificant contribution to Gamma).

Here we would like to point out that in Eq1 below G itself cannot be decomposed into G_{iav} and G_{clt} (same to Beta here), which however are estimated based on

decomposed interannual variation and climatology (20 year moving mean) of climate variables (Eq2). CGR_iav is thus derived following Eq3 as follows:

$$(E_{\text{clt}} + E_{\text{iav}}) - (CGR_{\text{clt}} + CGR_{\text{iav}}) = \mathbf{Beta} * (A_{\text{clt}} + A_{\text{iav}}) - (\mathbf{G}_{\text{clt}} + \mathbf{G}_{\text{iav}}) * (T_{\text{clt}} + T_{\text{iav}}) + O(\text{errors}) \quad (1)$$

$$(E_{\text{clt}} + E_{\text{iav}}) - (CGR_{\text{clt}} + CGR_{\text{iav}}) = \mathbf{Beta} * (A_{\text{clt}} + A_{\text{iav}}) - \mathbf{G} * (T_{\text{clt}} + T_{\text{iav}}) + O(\text{errors}) \quad (2)$$

$$CGR_{\text{iav}} = G_{\text{iav}} * T_{\text{iav}} + G_{\text{clt}} * T_{\text{clt}} + O(\text{errors}) \quad (3)$$

We thank reviewer he/she for providing suggestions for how to decompose and from the above description, we hope that the reviewer is able to follow our approach. We are happy to clarify further if needed be.

Fig. R1. Sensitivity of CGR to tropical temperature using method of M1 in our main manuscript and the method accounting for climatology (Eq3).

2) Biogeophysical/chemical interpretation: In the LPJ-GUESS model experiments, what carbon flux component mainly contributes to Gamma*T? Is it respiration (vegetation or/and soil carbon) or photosynthesis (GPP)? How exactly precipitation variations regulate these processes? Using only NBP cannot answer these questions.

Response: We thank reviewer for your insightful comment. We have now added an analysis of identifying the sensitivity of changes in each carbon flux component (i.e., NPP, R_h and fire emissions) to changes in the variations of CGR sensitivity (Gamma) using the simulated outputs from LPJ-GUESS.

By comparing changes in the temperature sensitivities of NPP ($\gamma_{\text{NPP}}^{\text{T}}$), R_h ($\gamma_{\text{Rh}}^{\text{T}}$) and fire emissions ($\gamma_{\text{FIRE}}^{\text{T}}$) with the sensitivity of NBP ($\gamma_{\text{NBP}}^{\text{T}}$), changes in $\gamma_{\text{NBP}}^{\text{T}}$ was found to be most sensitive to changes in $\gamma_{\text{NPP}}^{\text{T}}$, followed by $\gamma_{\text{Rh}}^{\text{T}}$ and $\gamma_{\text{FIRE}}^{\text{T}}$ (see Fig. R2a below). We further analyzed how precipitation regulates changes in $\gamma_{\text{NBP}}^{\text{T}}$ using the outputs from LPJ-GUESS. Changes in $\gamma_{\text{NBP}}^{\text{T}}$ were simulated under scenarios of SCE1 (see scenario in Extended Data Table 3) where all drivers of NBP are varying with time and SCE1-SCE2 where precipitation drives changes in NBP, respectively. We revealed a spatial

pattern of significant ($p < 0.05$) correlations between changes in $\gamma_{\text{NBP}}^{\text{T}}$, $\gamma_{\text{NPP}}^{\text{T}}$, $\gamma_{\text{Rh}}^{\text{T}}$ and $\gamma_{\text{FIRE}}^{\text{T}}$ and their changes driven by precipitation (Fig. R2b), which fully support our findings of the importance of precipitation in regulating $\gamma_{\text{CGR}}^{\text{T}}$.

We have now included these figures in the main manuscript as new Fig. 4 and Supplementary Fig. 16.

A new paragraph covering this was thus added (line 182-190):

“We further studied the sensitivity of changes in $\gamma_{\text{NPP}}^{\text{T}}$, $\gamma_{\text{Rh}}^{\text{T}}$, $\gamma_{\text{FIRE}}^{\text{T}}$ to $\gamma_{\text{NBP}}^{\text{T}}$ and found that changes in $\gamma_{\text{NBP}}^{\text{T}}$ was most sensitive to $\gamma_{\text{NPP}}^{\text{T}}$, followed by $\gamma_{\text{Rh}}^{\text{T}}$ and $\gamma_{\text{FIRE}}^{\text{T}}$ Fig. 4a) We further analyzed how precipitation regulates changes in the sensitivity of carbon fluxes to tropical temperature using the outputs from LPJ-GUESS. Changes in the sensitivity was simulated under scenarios of SCE1 (see scenario in Extended Data Table 3) where all drivers of carbon fluxes are varying with time and SCE1-SCE2 where precipitation drives changes, respectively. We revealed a widespread spatial pattern of significant ($P < 0.05$) correlations of $\gamma_{\text{NBP}}^{\text{T}}$, $\gamma_{\text{NPP}}^{\text{T}}$, $\gamma_{\text{Rh}}^{\text{T}}$, $\gamma_{\text{FIRE}}^{\text{T}}$ between SCE1 and SCE1-SCE2 (Fig.4b and Supplementary Fig. 16), which fully support our findings of the importance of precipitation in regulating changes in the sensitivity of carbon fluxes to temperature.”

Fig. R2. a, Changes in the sensitivity of changes in $\gamma_{\text{NPP}}^{\text{T}}$, $\gamma_{\text{Rh}}^{\text{T}}$ and $\gamma_{\text{FIRE}}^{\text{T}}$ to changes in $\gamma_{\text{NBP}}^{\text{T}}$. **b,** Spatial patterns of correlation coefficients of $\gamma_{\text{NBP}}^{\text{T}}$ under scenarios between SCE1 (all drivers vary with time) and SCE1-SCE2 (precipitation drives variation in NBP). The pixels with significant ($p < 0.05$) correlations were shown.

As such, I cannot recommend the publication of the paper on Nature Communications in its current form.

Response: We believe that we have now been able to address your concerns and suggestions which has allowed to substantially improved the manuscript. It has become more convincing. We thank you for that and hope you can now recommend the publication.

Reviewer #2 (Remarks to the Author):

This paper focused on the recent decrease in the sensitivity of the atmospheric CO₂ growth rate (CGR), and the authors combine data analysis and process-based model to show that the weaker sensitivity of CGR is controlled by increases in precipitation. CGR sensitivity has been extensively studied in the literature, including multiple high-profile papers. The finding regarding the recent decline in CGR sensitivity is new, and the authors did a good job analyzing and explaining the causes. Analyses methods used in this paper are mostly proven techniques reported in Wang et al. (2014) and Luo et al. (2022), and I have no issues regarding the robustness of the results. However, linkage between increased precipitation and decrease in CGR sensitivity could already be expected from previous research, so it would be better if new knowledge gained from analyses here could be better highlighted. While simulation results from process-based models were reported, it would make the paper stronger if they were analyzed further to advance our mechanistic understandings of the reported phenomenon.

Response: Thank you for your positive comments on this manuscript, the methods used and the robustness of the results. In the revised version of the manuscript, we have fully implemented your suggestions and thereby strengthened significance of the of the results by analyzing the sensitivities of each of the relevant carbon flux components to temperature (i.e., NPP, R_h and fire emission) using simulation data from LPJ-GUESS. This suggestion is in line with the comment 2 provided by reviewer#1; please see the response below for a detailed clarification of how the reviewer suggestion was addressed.

We have now added an analysis of identifying the sensitivity of changes in each carbon flux component (i.e., NPP, R_h and fire emissions) to changes in the variations of CGR sensitivity (Gamma) using the simulated outputs from LPJ-GUESS.

By comparing changes in the temperature sensitivities of NPP (γ_{NPP}^T), R_h (γ_{Rh}^T) and fire emissions (γ_{FIRE}^T) with the sensitivity of NBP (γ_{NBP}^T), changes in γ_{NBP}^T was found to be most sensitive to changes in γ_{NPP}^T , followed by γ_{Rh}^T and γ_{FIRE}^T (see Fig. R2a below). We further analyzed how precipitation regulates changes in γ_{NBP}^T using the outputs from LPJ-GUESS. Changes in γ_{NBP}^T were simulated under scenarios of SCE1 (see scenario in Extended Data Table 3) where all drivers of NBP are varying with time and SCE1-SCE2 where precipitation drives changes in NBP, respectively. We revealed a spatial pattern of significant ($p < 0.05$) correlations between changes in γ_{NBP}^T , γ_{NPP}^T , γ_{Rh}^T and γ_{FIRE}^T and their changes driven by precipitation (Fig. R2b), which fully support our findings of the importance of precipitation in regulating γ_{CGR}^T .

We have now included these figures in the main manuscript as new Fig. 4 and Supplementary Fig. 16.

A new paragraph covering this was thus added (line 182-190):

“We further studied the sensitivity of changes in γ_{NPP}^T , γ_{Rh}^T , γ_{FIRE}^T to γ_{NBP}^T and found that changes in γ_{NBP}^T was most sensitive to γ_{NPP}^T , followed by γ_{Rh}^T and γ_{FIRE}^T Fig. 4a) We further analyzed how precipitation regulates changes in the sensitivity of carbon fluxes to tropical temperature using the outputs from LPJ-GUESS. Changes in the sensitivity

was simulated under scenarios of SCE1 (see scenario in Extended Data Table 3) where all drivers of carbon fluxes are varying with time and SCE1-SCE2 where precipitation drives changes, respectively. We revealed a widespread spatial pattern of significant ($P < 0.05$) correlations of γ_{NBP}^T , γ_{NPP}^T , γ_{Rh}^T , γ_{FIRE}^T between SCE1 and SCE1-SCE2 (Fig. 4b and Supplementary Fig. 16), which fully support our findings of the importance of precipitation in regulating changes in the sensitivity of carbon fluxes to temperature.”

Fig. R2. a, Changes in the sensitivity of changes in γ_{NBP}^T , γ_{Rh}^T and γ_{FIRE}^T to changes in γ_{NBP}^T . **b**, Spatial patterns of correlation coefficients of γ_{NBP}^T under scenarios between SCE1 (all drivers vary with time) and SCE1-SCE2 (precipitation drives variation in NBP). The pixels with significant ($p < 0.05$) correlations were shown.

One area I would like to see improved is process understanding based on the simulation results. From the simulation results, LPJ-GUESS shows a clear signal of precipitation-driven changes in CGR sensitivity, but results from CMIP6 seem quite mixed. Is the mean sensitivity significantly different between the very wet and very dry conditions in Fig. 4a? In Fig. 4b, we see similar number of models above/below the 1:1 line. What might be the reasons for some CMIP6 models to exhibit quite different behavior compared to LPJ-GUESS model? Several factors such as the acclimation of terrestrial ecosystem to ongoing warming and increase in disturbances were mentioned, but since the acclimation effect is not represented in process-based models (is this right?), it is difficult to quantify its contribution to changes in CGR sensitivity. However, could disturbance be held fixed in one of the factorial experiments to analyze its role?

Response: Thank you for posing these questions. We also realized the insignificant ($p = 0.26$, tested by the two-sided Student’s t-test) differences in NBP sensitivity between very wet and very dry conditions with the results from CMIP6. Figure 5b in the revised manuscript shows the details regarding which ESMs can capture the observed differences in sensitivity under very wet and dry conditions. As we did not intend to run all CMIP6 models as explicit as LPJ-GUESS, the different behaviors from other ESMs compared to LPJ-GUESS could be only discussed. We think it could depend on

how they represent vegetation composition, nutrient limitation as well as drought impacts on plant photosynthesis. As far as we know, temperature acclimation is not represented in these models. In the revision, we have explicitly examined the contribution from fire emissions, one of the disturbance-related components most relevant in the tropics, and found that the temperature sensitivity of NBP is still mostly associated with NPP, indicating the key role of plant productivity in regulating the responses.

Secondly, while the paper in general is easy to follow, my impression is that the methods were not explained as clearly as in some other cited papers. For example, Luo et al. (2022) explained the Durbin-Watson indicator in detail, whereas this paper only mentioned the method was used. For readers familiar with the literature it is not an issue, as no new methodology was introduced. Other readers however, would have to look into multiple cited paper sources to understand fully some of the methodology. It would improve the readability if various techniques were better explained in this paper, similar to other papers.

Response: Thank you for this suggestion. In the revised manuscript, we have now added a more detailed description on how to calculate the Durbin-Watson index and how to obtain the adjusted variables (lines 422-435). “The Durbin-Watson indicator (DWI) was used to detect the autocorrelation at lag 1 in the residuals (e_i) from the ordinary linear regression of a target variable against time³⁹ using a significance level of $P < 0.001$ was set. DWI was calculated through the Eq. (3) as follows:

$$DWI = \frac{\sum_{i=2}^n (e_i - e_{i-1})^2}{\sum_{i=1}^n e_i^2} \quad 3$$

Where n is the number of observations, e_i is the i^{th} residual of the linear regression of the investigated variable against time. The DWI values range between 0 and 4. A value of 2 indicates no autocorrelation, whereas values from 0 to 2 indicate positive autocorrelation and values from 2 to 4 denote negative autocorrelation in the time series of observations.”

Lastly, unlike Luo et al. (2022), changes in water sensitivity of CGR or NBP were not reported in this paper. I wonder if additional insights could be gained from such analyses on the observed and simulated data.

Response: Thanks for bring up this interesting idea. Luo et al’s study has presented the changes in the sensitivity of water to CGR and an increase in the CGR sensitivity to water was in the last decades. This is a very interesting finding and several recent studies (refs 1-3) reported changes in the sensitivity of vegetation productivity (GPP/NDVI/LAI) to precipitation (or soil moisture), which can to some degree support the reported changes in the water sensitivity to CGR. However, we found large inconsistencies in the observed vs. the modelled CGR/NBP sensitivity to water, suggesting that the model cannot capture the temporal changes in the observed CGR sensitivity (Fig. 3 and TRENDY in Luo et al). This mismatch calls for improvements in process descriptions in models to realistically capture the response of carbon fluxes to varying water conditions.

Fig. R3. Changes in the sensitivity of CGR/NBP to water (referring to precipitation here)

Reference:

1 Zhang, Y., Gentile, P., Luo, X., Lian, X., Liu, Y., Zhou, S., Michalak, A.M., Sun, W., Fisher, J.B., Piao, S., Keenan, T.F., 2022. Increasing sensitivity of dryland vegetation greenness to precipitation due to rising atmospheric CO₂. *Nat. Commun.* 4875.
 2 Li, W., Migliavacca, M., Forkel, M., Denissen, J.M.C., Reichstein, M., Yang, H., Duveiller, G., Weber, U., Orth, R., 2022. Widespread increasing vegetation sensitivity to soil moisture. *Nat. Commun.* 1–9.
 3 Zeng, X., Hu, Z., Chen, A., Yuan, W., Hou, G., Han, D., Liang, M., Di, K., Cao, R., Luo, D., 2022. The global decline in the sensitivity of vegetation productivity to precipitation from 2001-2018. *Glob. Chang. Biol.*

Some minor grammatical issues:

P1L37: “most challenging tasks” to simulate correctly?

Response: We have rephrased this sentence as “the sensitivity of tropical carbon sequestration to climate change remains a challenging task for Earth system models (ESMs) when predicting future climate change”.

P8L218: “reversal”-> “a reversal”

Response: Thanks. It has been revised accordingly.

P9L252: “evidence” -> “provide evidence”

Response: Done.

Reviewer #3 (Remarks to the Author):

Review of “Recent decrease of the impact of tropical temperature on the carbon cycle linked to increased precipitation”, by Zhang et al., submitted to Nature Communications (Manuscript #370727)

This is an interesting paper that diagnoses changes in the sensitivity of the growth-rate of atmospheric CO₂ to tropical temperature variations (γ_{CGR}), and presents evidence that these are due to changing moisture availability in the tropics. It follows on from previous Nature papers which (a) identified that this metric is well correlated with projected carbon loss in the tropics due to future warming (Cox et al., 2013); and (b) showed evidence that CGR had increased since 1960 (Wang et al., 2014).

This Zhang et al. submission suggests that γ_{CGR} (as calculated within 20 year windows) has decreased since 1990 and is now back close to its 1970 level. The reason for this decline is traced to a minimum in tropical precipitation around 1990. The paper is relevant to climate change and climate variability and its impact on tropical land carbon stores. It is therefore potentially of broad general interest to the readership of Nature Communications.

Response: We thank the reviewer for his/her positive evaluation of the interest and relevancy of our manuscript.

However, I think the current draft needs to be improved in a number of ways before it should be published:

1) Mechanisms: the paper currently offers little explanation for the changes in \square CGR aside from the correlation with overall tropical mean precipitation (and the related PDSI). What are the possible mechanisms that would lead to a dependency of the sensitivity of γ_{CGR} ? How could you distinguish between these possible mechanisms? Wouldn't you expect γ_{CGR} to be roughly proportional to GPP? If so, shouldn't γ_{CGR} increase (rather than decrease) with less drought-limited GPP? How does your suggestion of reducing moisture stress in the tropics compare to other recent papers suggesting drying and reducing resilience of the Amazon rainforest (e.g. Boulton et al., 2022)? These issues need to be discussed somewhere in a revised paper – ideally in the Discussion section which is currently a little repetitive;

Response: Thanks for these constructive suggestions. We have now analysed and added the changes in the sensitivity of the most relevant carbon fluxes (i.e., NPP, R_h and fire emission) to temperature to further link them to the changes in CGR sensitivity. Please see Fig. R2 below. The spatial patterns regarding the relationships between carbon fluxes and precipitation were presented as suggested by the reviewer #1 (see more detailed comments in the following paragraph). We believe that these revisions greatly helped us to reveal the mechanisms of changes in γ_{CGR}^T . From the observations presented in our study, γ_{CGR}^T is positively related to decreased water availability, which also agreed with previous findings (ref 1-2).

1. Wang, X., Piao, S., Ciais, P., Friedlingstein, P., Myneni, R.B., Cox, P., Heimann, M., Miller, J., Peng, S., Wang, T., Yang, H., Chen, A., 2014. A two-fold increase of carbon cycle sensitivity to tropical temperature variations. Nature 506, 212–215.

2.Luo, X., Keenan, T.F., 2022. Tropical extreme droughts drive long-term increase in atmospheric CO2 growth rate variability. Nat. Commun. 13, 1193.

Fig. R2. a, Changes in the sensitivity of changes in γ_{NBP}^T , γ_{Rh}^T and γ_{FIRE}^T to changes in γ_{NBP}^T . **b**, Spatial patterns of correlation coefficients of γ_{NBP}^T under scenarios between SCE1 (all drivers vary with time) and SCE1-SCE2 (precipitation drives variation in NBP). The pixels with significant ($p < 0.05$) correlations were shown.

An overall decrease in vegetation resilience that was observed in the Amazon rainforest (ref 3) is an interesting study. However, Boulton et al' study did not attribute the decrease in resilience to drying, as they found no significant correlation between changes in precipitation and changes in resilience (Supplementary Figure S12 in Boulton et al paper). Moreover, the temporal span for the finding in ref. (3) is limited to 1991-2016, which is shorter than the observation period presented in our study (1960-2020). The changes in moisture stress could thus be different. We calculated precipitation using a 20-year moving window, for a long-term at a bi-decadal scale, and decreasing moisture stress was observed in the last decade (Extended Data Fig. 12a-b).

We have now discussed (lines 262-264) the mechanisms behind the detected changes in CGR sensitivity in the revised discussion section. “Lastly, the simulation results show that water dryness controlling changes in γ_{NBP}^T were primarily determined by γ_{NPP}^T , followed by γ_{Rh}^T and γ_{FIRE}^T , suggesting that plant productivity is still key in regulating γ_{CGR}^T ”

3.Boulton, C.A., Lenton, T.M. & Boers, N. Pronounced loss of Amazon rainforest resilience since the early 2000s. Nat. Clim. Chang. 12, 271–278 (2022).

2) **Spatial Detail:** one way to separate out the mechanisms is to look not just at means for the whole tropics, but instead to look also at the spatiotemporal variability of the climate and how this is correlated with the spatiotemporal variability of net carbon uptake. This is obviously difficult from observations alone (because we do not have measurements of carbon uptake for each gridpoint in the real world), but it can be done with models. I notice also that there are maps in the Supplementary Material. Some explanatory spatial detail (maps etc.) needs to be included in the main manuscript;

Response: Thanks for this insightful suggestion. We have now added the maps showing the spatial correlations between the NBP sensitivities derived from model and bi-decadal precipitation in our main manuscript (see Fig. R2).

A new paragraph was thus added in line 182-190: “*We further studied the sensitivity of changes in $\gamma_{\text{NPP}}^{\text{T}}$, $\gamma_{\text{Rh}}^{\text{T}}$, $\gamma_{\text{FIRE}}^{\text{T}}$ to $\gamma_{\text{NBP}}^{\text{T}}$ and found that changes in $\gamma_{\text{NBP}}^{\text{T}}$ was most sensitive to $\gamma_{\text{NPP}}^{\text{T}}$, followed by $\gamma_{\text{Rh}}^{\text{T}}$ and $\gamma_{\text{FIRE}}^{\text{T}}$ Fig. 4a) We further analyzed how precipitation regulates changes in the sensitivity of carbon fluxes to tropical temperature using the outputs from LPJ-GUESS. Changes in the sensitivity was simulated under scenarios of SCE1 (see scenario in Extended Data Table 3) where all drivers of carbon fluxes are varying with time and SCE1-SCE2 where precipitation drives changes, respectively. We revealed a widespread spatial pattern of significant ($P < 0.05$) correlations of $\gamma_{\text{NBP}}^{\text{T}}$, $\gamma_{\text{NPP}}^{\text{T}}$, $\gamma_{\text{Rh}}^{\text{T}}$, $\gamma_{\text{FIRE}}^{\text{T}}$ between SCE1 and SCE1-SCE2 (Fig.4b and Supplementary Fig. 16), which fully support our findings of the importance of precipitation in regulating changes in the sensitivity of carbon fluxes to temperature.”*

3) **DGVM Modelling results:** it is good to see an attempt at corroboration with model results, although this is currently inconclusive. A single DGVM model (LPJ-GUESS) is used, but we know that models differ in their responses to climate variations (with no obvious best model). I advise the authors to instead use data from each of the Trendy models;

Response: Thanks for this suggestion. We have now included results of the changes in NBP sensitivities using multiple TRENDY DGVMs, which generally agreed well with the observations from LPJ-GUESS model (Fig. 4). DGVMs in the TRENDY project run under a protocol that did not allow to analyze the modulation of $\gamma_{\text{NBP}}^{\text{T}}$ by the IAV and trends of precipitation, thus preventing us to examine impacts of each individual climate variables (e.g., precipitation, temperature) and identifying the relative importance of different processes. We thus chose to place the newly-added analysis and results of DGVMs in the supplementary material to support our findings (Figure S15).

Fig. R4. Changes in the average sensitivities of NBP to tropical temperature for the period 1960–2020. The NBP sensitivities were computed with CRU annual climatic data. DGVMs included CLASSIC, CLM5.0, DLEM, IBIS, ISBA-CTRIP, JSBACH, JULE-ES, LPJ-GUESS, LPX-Bern, OCN, ORCHIDEE-CNP, ORCHIDEEv3, SDGVM, VISIT, YIBs. The outputs under scenario three for each model were used in this study. The grey shading represents the mean \pm 1 s.d.

4) CMIP6 Model results: the CMIP6 models shown in Figure 4 show no significant difference between γ_{CGR} for ‘very wet’ and ‘very dry’ years (note overlapping uncertainty bars in Figure 4a), although the manuscript underplays this (“These results indicated that ESMs can to some extent capture the response of the carbon flux to changes in temperature under very wet and dry conditions”). The comparison between models and observations needs to be much clearer, e.g. by including equivalent observational estimates on Figure 4(a) and 4(b);

Response: Thanks. We have noticed the insignificant differences in γ_{NBP}^T for wet and dry years ($p = 0.26$, based on the two-sided Student’s t-test). We have included the observed γ_{NBP}^T in Figure 4b which now corresponds to Figure 5b in the revised manuscript labeled by A in red color to better allow a direct comparison with the modelled values. For this figure, each point indicates one observation from either a station or model. Figure 4a is the ensemble mean γ_{NBP}^T for 27 ESMs, which can be used to calculate the standard deviations indicating a confidence interval. We thus chose not to add the observation of CGR sensitivity on Figure 4a (Figure 5b in the revised manuscript).

Also, we have modified the sentence mentioned by reviewer slightly towards "can only to some extent capture the response"

5) Clarity of model data comparisons: in general the model data comparisons are not made very clearly. I advise the authors to include their observational estimates on Figures 3(a) and 4, and to include a best fit straightline on Figure 4(b) in addition to the one-to-one line. The points in Figure 4(b) are also very difficult to see because of the long model names on the plot. Please remove these model names and instead include distinguishable symbols (e.g. ‘a’, ‘b’, ‘c’ etc.) and a key;

Response: Thanks for the suggestions, which were implemented accordingly. The observation-based γ_{CGR} were added in the revised manuscript (see OBS in Fig. 5 below). The best fitted line was as well added on Figure 4b (Figure 5b in the revised

manuscript) (Fig. 6 below) and the ESMS names were replaced for a letter for each model (see Supplementary table 4).

Fig. R5. Changes in the sensitivity of NBP to temperature under different simulations for the period 1960–2020. For the details of simulations is referred to Supplementary Table 3. OBS denotes the observed CGR sensitivity to tropical temperature.

Fig. R6. Sensitivity of the tropical terrestrial carbon sink to temperature simulated in the CMIP6 models. The red line is the 1:1 line and the blue line is the best fit line in Fig. R6b.

6) Clarity of the language: in general the paper is well written but there are some typos that need correcting. Given the other more significant revisions listed above, I don't intend to go through these one by one now, but will do so if I am asked to review a revised manuscript.

Response: Thanks for pointing this out and thanks for your very kind offer. The manuscript has now been read by all my co-authors and we have tried our best to correct the typos of the previous version.

Overall, I think this is an interesting study, and is of appropriately broad interest for Nature Comm., but needs these fairly major revisions before publication.

Response: Thank you for your encouragement and valuable feedback. Thanks also for having propitiated a more robust and interesting paper with your suggestions for improvement.

REVIEWER COMMENTS

Reviewer #1 (Remarks to the Author):

Comments to “Recent decrease of the impact of tropical temperature on the carbon cycle linked to increased precipitation” by Wenmin Zhang (revised version)

I truly appreciate the authors’ responses to my previous comments but found these responses did not fully address my concern. In particular, my original comment #1 suggested that the CGR sensitivity to temperature, Gamma or G for brevity, should include a term $G_{iav} * T_{clt}$ (which was missing in the statistical model used by the paper):

$$CGR_{iav} = G * T_{iav} + G_{iav} * T_{clt} + O(errors)$$

The authors’ response states that this term is insignificant because this term does not change the estimate of Gamma or G:

Fig. R1. Sensitivity of CGR to tropical temperature using method of M1 in our main manuscript and the method accounting for climatology (Eq3).

Their response, however, missed my point. What I really suggested was that the authors should do a first-order magnitude check for their results based on the suggested equation:

Assume the G_{iav} estimated by the paper is correct (as shown Fig.R1 above), we see that G_{iav} is roughly about $0.2 \text{ PgC} * \text{yr}^{-2} * \text{K}^{-1}$ between 1970-1990 and changed to about $-0.1 \text{ PgC} * \text{yr}^{-2} * \text{K}^{-1}$ from 1990 to 2010 (also see Fig. 1 in the manuscript, which is referred to as the trend of G). Assume the mean tropical air temperature is $20 \text{ }^\circ\text{C}$ (I highly doubted that the authors really used Kelvin degrees in their analysis, which will induce a much larger error), we then see that the term $G_{iav} * T_{clt}$ leads to an increase of CGR by $4 \text{ PgC} * \text{yr}^{-2}$ from 1970 to 1990 and an decrease of CGR by $-2 \text{ PgC} * \text{yr}^{-2}$ from 1990 to 2010, which are obviously incorrect! This contradicts the initial assumption – in another word, the estimated G_{iav} is incorrect.

I am sorry to be the bad guy here. But I really can’t let this kind of mistakes pass by...

[p.s, my original comment #1 and the authors' responses]

My specific comments/questions are as follows:

1) The statistical models used in this study, as suggested by Eq. 1 or Eq. 2, are not complete but neglect some important terms. From the point of global carbon cycling it is quite straightforward to see that

$$C \text{ Sink} = \text{Emission} - \text{CGR} = \text{Beta} * A_{\text{co2}} - \text{Gamma} * T + O(\text{errors})$$

where Beta is the CO2 fertilization effect, A_{co2} is the atmospheric CO2 concentration, and Gamma (or G) is the CGR sensitivity to T. We can decompose the variables to the climatology (~20yr moving mean) and the interannual variations as

$$(E_{\text{clt}} + E_{\text{iav}}) - (\text{CGR}_{\text{clt}} + \text{CGR}_{\text{iav}}) = \text{Beta} * (A_{\text{clt}} + A_{\text{iav}}) - (G_{\text{clt}} + G_{\text{iav}}) * (T_{\text{clt}} + T_{\text{iav}}) + O(\text{errors})$$

Note that because Gamma (G) is not a constant, it has to be decomposed to climatological and interannual components as temperature. Subtracting climatologies terms and neglecting smaller variables (e.g., E_{iav} and A_{iav}) leads to

$$\text{CGR}_{\text{iav}} = G_{\text{clt}} * T_{\text{iav}} + G_{\text{iav}} * T_{\text{iav}} + G_{\text{iav}} * T_{\text{clt}} + O(\text{errors}) = G * T_{\text{iav}} + G_{\text{iav}} * T_{\text{clt}} + O(\text{errors})$$

So we see the statistical models of Eq. 1 or 2 missed, at least, the $G_{\text{iav}} * T_{\text{clt}}$ term. The incompleteness of the model could contribute to the over-/under-estimation of G_{iav} .

Response: Thanks for this comment. For the response, we have decomposed these variables (climate and CGR) into interannual variations and climatology (20 year moving mean) to calculate CGR sensitivities in response to short-term (annual) and long-term (20-year) changes. We found no statistically significant difference ($p = 0.8962$) in $\text{Gamma}_{\text{iav}}$ (γ_{CGR}^T , calculated sensitivity) calculated by our initial approach and the approach proposed by the reviewer, based on the two-sided Student's t-test and the result is shown below in the Fig. R1. (R denotes figures presented in the response letter) (M1 denote the $\text{Gamma}_{\text{iav}}$ calculated using Eq 1 in our manuscript and the line called "Account for climatology" show the result of the calculation after accounting for climatology). To be consistent with previous publications on the subject, we therefore propose not to separate climatology + interannual variations when calculating $\text{Gamma}_{\text{iav}}$ (due to its insignificant contribution to Gamma).

Here we would like to point out that in Eq1 below G itself cannot be decomposed into G_{iav} and G_{clt} (same to Beta here), which however are estimated based on decomposed interannual variation and climatology (20 year moving mean) of climate variables (Eq2). CGR_{iav} is thus derived following Eq3 as follows:

$$(E_{\text{clt}} + E_{\text{iav}}) - (\text{CGR}_{\text{clt}} + \text{CGR}_{\text{iav}}) = \text{Beta} * (A_{\text{clt}} + A_{\text{iav}}) - (G_{\text{clt}} + G_{\text{iav}}) * (T_{\text{clt}} + T_{\text{iav}}) + O(\text{errors}) \quad (1)$$

$$(E_{\text{clt}} + E_{\text{iav}}) - (\text{CGR}_{\text{clt}} + \text{CGR}_{\text{iav}}) = \text{Beta} * (A_{\text{clt}} + A_{\text{iav}}) - G * (T_{\text{clt}} + T_{\text{iav}}) + O(\text{errors}) \quad (2)$$

$$\text{CGR}_{\text{iav}} = G_{\text{iav}} * T_{\text{iav}} + G_{\text{clt}} * T_{\text{clt}} + O(\text{errors}) \quad (3)$$

We thank reviewer he/she for providing suggestions for how to decompose and from the above description, we hope that the reviewer is able to follow our approach. We are happy to clarify further if needed be.

(Fig. R1 and Caption here)

Reviewer #2 (Remarks to the Author):

The revised manuscript has greatly improved on clarity of figures and methodology. However, some response to reviewers' questions is not fully satisfactory. After further reading, it also seems that some of the results, while robust, do not fully support the conclusions. Therefore, I advise the authors to directly address some of the harder questions, and to tone down some conclusions. Specific points are below:

1. I notice that all reviewers have commented on lack of discussion on mechanisms. While the revised version now added changes in sensitivity of some carbon fluxes, it is still the same correlation analysis which does not imply causation. I feel some questions from reviewer 3 are not answered directly (especially those related with GPP). It would help if some discussion on GPP is included in the revised paper.
2. After seeing the updated Fig. 3a, Fig. R3 and R4, my impression is that the LPJ-GUESS model has some difficulty capturing observed CGR sensitivity change (peak a bit too early, a bit too low after 2000, and change in the sensitivity of CGR/NBP to water differ from observation), which casts doubt on conclusions based on results from the single DGVM. If other drivers for the decreased sensitivity are not implemented well in the DGVM, it would affect analyses in Fig. 3, and the model could simulate the right magnitude of NBP sensitivity change for wrong reasons. I advise the authors to include some discussions related to Fig. R3 in the main paper, and to tone down some conclusions (for example, one could say instead "implying that increases in precipitation is a driver for the decreased CGR sensitivity during recent decades").
3. The added Fig. 4b in my view is not the most effective way to convey spatial details as the colors look quite similar (the map does not provide much added value compared to the inset). It would be much more interesting if a spatial figure of NBP sensitivity contribution from different regions is shown instead, perhaps by LPJ-GUESS and other TRENDY models.

Reviewer #3 (Remarks to the Author):

I thank the authors for their diligent attention to my comments. I am happy with most of what they have done. However, there are still some things that need attention:

- 1) The new figure 4 is useful (even though it is only with one DGVM), but the paper does not define what is meant by "sensitivity" here. Is this the regression coefficient for each factor on γ^T_{NBP} , or something else? Needs explaining.
- 2) As I said previously, the language of the paper needs attention from the co-authors (or perhaps the editor). There are numerous examples where the wrong tense is used, or where statements are made with undue definitiveness. To list just a few:
 - a) In the abstract: "...but it is unknown if the increase is stationary" - doesn't make much sense. Better to say "but this trend has not continued".
 - b) "suggesting" rather than "implying".
 - c) Line 56: "Here, we investigated" should read *investigate*.
 - d) Results should be written in the present sense.Basically, the paper now needs to have the English improved somewhat throughout (the co-authors might need to take some responsibility here..;-).

Reviewer #1 (Remarks to the Author):

Comments to “Recent decrease of the impact of tropical temperature on the carbon cycle linked to increased precipitation” by Wenmin Zhang (revised version)

I truly appreciate the authors’ responses to my previous comments but found these responses did not fully address my concern. In particular, my original comment #1 suggested that the CGR sensitivity to temperature, Gamma or G for brevity, should include a term $G_{iav} * T_{clt}$ (which was missing in the statistical model used by the paper):

$$CGR_{iav} = G * T_{iav} + G_{iav} * T_{clt} + O(errors)$$

The authors’ response states that this term is insignificant because this term does not change the estimate of Gamma or G:

Their response, however, missed my point. What I really suggested was that the authors should do a first-order magnitude check for their results based on the suggested equation:

Assume the G_{iav} estimated by the paper is correct (as shown Fig.R1 above), we see that G_{iav} is roughly about $0.2 \text{ PgC} * \text{yr}^{-2} * \text{K}^{-1}$ between 1970-1990 and changed to about $-0.1 \text{ PgC} * \text{yr}^{-2} * \text{K}^{-1}$ from 1990 to 2010 (also see Fig. 1 in the manuscript, which is referred to as the trend of G). Assume the mean tropical air temperature is $20 \text{ }^\circ\text{C}$ (I highly doubted that the authors really used Kelvin degrees in their analysis, which will induce a much larger error), we then see that the term $G_{iav} * T_{clt}$ leads to an increase of CGR by $4 \text{ PgC} * \text{yr}^{-2}$ from 1970 to 1990 and an decrease of CGR by $-2 \text{ PgC} * \text{yr}^{-2}$ from 1990 to 2010, which are obviously incorrect! This contradicts the initial assumption – in another word, the estimated G_{iav} is incorrect.

I am sorry to be the bad guy here. But I really can’t let this kind of mistakes pass by...[p.s, my original comment #1 and the authors’ responses]

Response: We thank the reviewer for providing further explanation, and realize that we indeed have misunderstood the earlier point.

The temperature data (as well as other data) were detrended prior to use in the analysis, and in this process, the offset is removed, resulting in a time series that varies around 0. Hence, TMP in Equation 1

$$CGR = \gamma_{CGR}(TMP) + \tau_{CGR}(PR) + \delta_{CGR}(SR) + \varepsilon$$

is not the temperature, but the temperature variation. Hence, T_{clt} in the reviewer’s equation is 0 by default, so the term disappears (and as another result of this, the use of K or $^\circ\text{C}$ does not matter).

We have clarified the description in the methods section by adding this sentence:

“All variables were detrended, resulting in anomalies of temperature (ΔT), precipitation (ΔP) and incoming shortwave radiation (ΔR) relative to their long-term linear trend. These anomalies were used to calculate the slope for a 20-year moving window between 1960–2020”

$$\text{CGR} = \gamma_{\text{CGR}}^T \Delta T + \tau_{\text{CGR}}^P \Delta P + \delta_{\text{CGR}}^R \Delta R + \varepsilon$$

Moreover, we observe a general agreement between changes in G in Fig. R1a and G_iav our manuscript in Fig. 1b, which suggest that both methods are consistent in calculating G/G_iav. Please note that the slight differences between them can be attributed to the fact that for the calculation based on this equation suggested we did not include the covarying variables of precipitation and solar radiation in the calculation, and did not exclude the years of 1992-1993 of Mt Pinatubo eruption.

Fig. R1 Changes in G according to the reviewer suggestion. This analysis is calculated according to the reviewer’s suggestion as follows:

$$\text{CGR_iav} = G * T_iav + G_iav * T_clt + O(\text{errors}).$$

Here, T_iav: detrended temperature; T_clt: 20-yr moving mean temperature

Reviewer #2 (Remarks to the Author):

The revised manuscript has greatly improved on clarity of figures and methodology. However, some response to reviewers' questions is not fully satisfactory. After further reading, it also seems that some of the results, while robust, do not fully support the conclusions. Therefore, I advise the authors to directly address some of the harder questions, and to tone down some conclusions. Specific points are below:

1. I notice that all reviewers have commented on lack of discussion on mechanisms. While the revised version now added changes in sensitivity of some carbon fluxes, it is still the same correlation analysis which does not imply causation. I feel some questions from reviewer 3 are not answered directly (especially those related with GPP). It would help if some discussion on GPP is included in the revised paper.

Response: Thank you for this suggestion. We fully agree that the correlation does not imply causation. We have included some additional discussion on the mechanisms involving photosynthesis (GPP) and the associated carbon components as follows.

“The decreased $\gamma_{\text{CGR}}^{\text{T}}$ is considered plausible, as alleviated water stress observed in recent decades could promote photosynthesis that sequesters more atmospheric CO_2 . These findings suggest that it is the interactions between water and temperature that control the carbon cycle in the tropical regions. Lastly, the simulation results show that water dryness controlling changes in $\gamma_{\text{NBP}}^{\text{T}}$ are primarily determined by $\gamma_{\text{NPP}}^{\text{T}}$, followed by $\gamma_{\text{Rh}}^{\text{T}}$ and $\gamma_{\text{FIRE}}^{\text{T}}$, suggesting that plant productivity is still key in regulating $\gamma_{\text{CGR}}^{\text{T}}$. Still, model limitations exist in capturing the dynamics of $\gamma_{\text{CGR}}^{\text{T}}$ across the tropics, which could thereby lead to uncertainties in the assessment of primary drivers for $\gamma_{\text{CGR}}^{\text{T}}$.”
Lines 270-277.

2. After seeing the updated Fig. 3a, Fig. R3 and R4, my impression is that the LPJ-GUESS model has some difficulty capturing observed CGR sensitivity change (peak a bit too early, a bit too low after 2000, and change in the sensitivity of CGR/NBP to water differ from observation), which casts doubt on conclusions based on results from the single DGVM. If other drivers for the decreased sensitivity are not implemented well in the DGVM, it would affect analyses in Fig. 3, and the model could simulate the right magnitude of NBP sensitivity change for wrong reasons. I advise the authors to include some discussions related to Fig. R3 in the main paper, and to tone down some conclusions (for example, one could say instead “implying that increases in precipitation is a driver for the decreased CGR sensitivity during recent decades”).

Response: Thanks for pointing out this. Yes, we agree that the model simulation of changes in CGR sensitivity is not fully replicating the observations, and consequently we modified the statement in the abstract as suggested by reviewer and we added the following statements in the main text:

“collectively suggesting that increases in precipitation control the decreased $\gamma_{\text{CGR}}^{\text{T}}$ during recent decades.” in the abstract.

“Overall, $\gamma_{\text{CGR}}^{\text{T}}$ shows a stronger variation during recent years, and the simulated decline in $\gamma_{\text{NBP}}^{\text{T}}$ starts earlier than the decline in observed $\gamma_{\text{CGR}}^{\text{T}}$ (Fig. 3a). However, the general agreement suggests that the interannual variation of CGR is primarily driven by the tropical land carbon flux (NBP)”. Lines 168-172.

We further added the following text in the discussion section, pointing out the uncertainty on the conclusions based on the model simulations.

“Still, model limitations exist in capturing the dynamics of $\gamma_{\text{CGR}}^{\text{T}}$ across the tropics, which could thereby lead to uncertainties in the assessment of primary drivers for $\gamma_{\text{CGR}}^{\text{T}}$.” Lines 275-277.

3. The added Fig. 4b in my view is not the most effective way to convey spatial details as the colors look quite similar (the map does not provide much added value compared to the inset). It would be much more interesting if a spatial figure of NBP sensitivity contribution from different regions is shown instead, perhaps by LPJ-GUESS and other TRENDY models.

Response: Thanks for pointing out this and for the relevant suggestion. We now have improved the color scale to make it clearer (Fig. R2 below and Fig. 4 in the revised manuscript).

As for the suggestion, we have produced a new figure showing the respective contributions of different tropical regions to changes in $\gamma_{\text{NBP}}^{\text{T}}$. We fully agree that this new figure is informative, and we have chosen to include this figure in the supplementary material as Figure S16 (Fig. R3 below). The following text were inserted to introduce the figure:

“In addition, according to the simulations of LPJ-GUESS, we find that that the relative contribution to changes in tropical $\gamma_{\text{NBP}}^{\text{T}}$ from the tropical regions of the different continents are dominated by tropical Africa, followed by Asia-Australia and South America (Supplementary Fig. 16)” Lines 576-178.

Fig. R2 Carbon flux component sensitivity to tropical temperature in relation to precipitation as simulated in LPJ-GUESS.

Fig. R3 Contribution to changes in γ_{NBP}^T from different tropical regions. Relative importance of the tropical regions of different continents (Africa (AF), Asia-Australia (AA) and South America (SA)) to the changes in γ_{NBP}^T for the entire tropical region. Results are based on a multiple linear regression model with γ_{NBP}^T of the entire tropical region used as a response variable and γ_{NBP}^T of the different regions used as explanatory variables. The explanatory power is 95% and the relative importance is assessed using the “lmg” approach, which is based on sequential R^2 but accounts for

the dependence on ordering of explanatory variables. Error bars denoted 1 SD of the explanatory power in 500 bootstrap estimates.

Reviewer #3 (Remarks to the Author):

I thank the authors for their diligent attention to my comments. I am happy with most of what they have done. However, there are still some things that need attention:

1) The new figure 4 is useful (even though it is only with one DGVM), but the paper does not define what is meant by "sensitivity" here. Is this the regression coefficient for each factor on γ_{NBP}^T , or something else? Needs explaining.

Response: Thanks for pointing out this. We have made this clearer now, by inserting the following sentence in the figure caption "The sensitivity is calculated as the regression coefficients for each explanatory variable in a multiple linear regression with the response variable being trends in γ_{NBP}^T and the explanatory variables being trends in γ_{NPP}^T , γ_{Rh}^T , and γ_{FIRE}^T ".

2) As I said previously, the language of the paper needs attention from the co-authors (or perhaps the editor). There are numerous examples where the wrong tense is used, or where statements are made with undue definitiveness. To list just a few:

- a) In the abstract: "..but it is unknown if the increase is stationary" - doesn't make much sense. Better to say "but this trend has not continued".
- b) "suggesting" rather than "implying".
- c) Line 56: "Here, we investigated" should read *investigate*.
- d) Results should be written in the present tense.

Basically, the paper now needs to have the English improved somewhat throughout (the co-authors might need to take some responsibility here.;-).

Response: We thank the reviewer for spotting and suggesting these revisions, which have all been implemented. We have now used the present tense throughout the results section, accordingly.

Co-authors have read through the manuscript again and corrected several instances of typos/grammar in the previous version of the manuscript.

REVIEWER COMMENTS

Reviewer #1 (Remarks to the Author):

It seems that the authors still didn't understand my critique. So I will do it one last time and try my best to clarify my points. I will go back to my comments of the first round as there turned out to be a mistake in my second-round comments, which should be dismissed – and I apologize for it.

Let me start by saying that the *variations or anomalies of a physical variable (e.g., temperature) do not exist independently by themselves*. If we calculate anomalies without a clear idea of the physics, we could introduce errors. Therefore, we need to be extra careful whenever we do additional manipulations (including detrending, masking, etc.) to the data. Verify the results with the original simple time series as much as you can.

To illustrate this point, let us return to the original equation that describes the dynamics of atmospheric CO2 growth rate:

$$\dot{E} - \dot{C} = f(C, T, \dots) \quad (1)$$

where \dot{E} is the annual CO2 emission rate, \dot{C} is the annual CO2 growth rate, C is the atmospheric CO2 level, and T is temperature. Other variables such as precipitation (P) and radiation (R) are implicitly included in the dynamic system. $\dot{E} - \dot{C}$ represents the annual carbon sink and the physical meaning of the equation is very clear.

The function f is nonlinear and can be linearized around a point $X = (C_X, T_X, \dots)$

$$\dot{E} - \dot{C} = f(X) + \beta_X \cdot C'_X + \gamma_X \cdot T'_X + \dots \quad (2)$$

where the sensitivity coefficients are defined by the Jacobians at the point X . The term $f(X)$ is generally non-zero unless X is an equilibrium point of the system, such that

$$f(X) = 0 \quad (3)$$

We may assume that the climatology of variables at the beginning of the studying period (1950s-1960s) represents a quasi-equilibrium, such that

$$\dot{E} - \dot{C} = \beta_X \cdot C'_X + \gamma_X \cdot T'_X + \dots \quad (4)$$

(The climatologies in each of the 20 years moving window are not equilibrium, and therefore a constant term should be included in the regression equation. It is a better practice than removing the means of the variables because we can then check if the means satisfy the same equation.)

Next the paper “detrended” the time series. This is not a desirable practice, but if we do want to separate the long-term (LT) and the interannual (IA) variations of the time series, we must observe certain constraints. As described in my previous comment, we decompose the terms in Eq. (4) into LT and IA components (the LT components are not necessarily a linear trend, but this is trivial and I won't distinguish the terms).

$$\dot{E}_{LT} - \dot{C}_{LT} - \dot{C}_{IA} = \beta \cdot C'_{LT} + (\gamma_{LT} + \gamma_{IA}) \cdot (T'_{LT} + T'_{IA}) + \dots \quad (5)$$

where we neglected the interannual component of CO2 emissions (\dot{E}_{IA}) and atmospheric CO2 concentration (\dot{C}'_{IA}) for they are generally small. Because this study focuses on the changes of CGR's sensitivity on temperature (γ), we also decompose it as well. Now we can separate the long-term trend and the interannual variations as follows:

$$\dot{E}_{LT} - \dot{C}_{LT} = \beta \cdot C'_{LT} + \gamma_{LT} \cdot T'_{LT} + \dots \quad (6)$$

and

$$\dot{C}_{IA} = -(\gamma_{LT} + \gamma_{IA}) \cdot T'_{IA} + \gamma_{IA} \cdot T'_{LT} + \dots \quad (7)$$

(

The paper separates the two components in a different way as

$$\dot{E}_{LT} - \dot{C}_{LT} = \beta \cdot C'_{LT} + (\bar{\gamma} + \gamma') \cdot T'_{LT} + \dots \quad (6^*)$$

$$\dot{C}_{IA} = -(\bar{\gamma} + \gamma') \cdot T'_{IA} + \dots \quad (7^*)$$

This treatment is less optimal because it leaves the IA variations $\gamma_{IA} \cdot T'_{LT}$ unaccountable.

)

It is clear that

- *The long-term component of T'_{LT} is certainly not zero and it has important impacts on \dot{C}_{IA} with the sensitivity γ_{IA} .* This term is missing from the model used in the paper because the time series were detrended and, most likely, have the means removed.

(This responds to the authors' rebuttal that says "The temperature data (as well as other data) were detrended prior to use in the analysis, and in this process, the offset is removed, resulting in a time series that varies around 0... Hense, T_clt in the reviewer's equation is 0 by default, so the term disappears")

- More critically, because the paper detrended the data, Eq. (6) totally disappears in the analysis. We really need Eq. (6) to verify the results reported in the paper because both Eqs. (6) and (7) must be simultaneously satisfied so that Eq. 5 is satisfied. This verification is important because, again, the interannual variations or anomalies of a physical variable (e.g., temperature) are pretty much an artificial concept and do not exist independently by themselves.

Therefore, what I asked the authors to do in my first-round comments were:

1. Estimate $\bar{\gamma}$ and γ' using Eq. (7);
2. Estimate β using Eq. (6) and the results ($\bar{\gamma}$ and γ') from the preceding step. Check if the estimates of β (CO2 fertilization effect) comparable to those reported recently.
3. Predict \dot{C} (i.e., $\dot{C}_{LT} + \dot{C}_{IA}$) using Eq. (5) and compare the results to the observations.

Finally, I also want to read more discussions on the exclusion of Pinatubo eruption era (1992-1993) from the data, in particular,

4. What will happen if you don't exclude this time period from the analysis?
5. I don't think extra cooling is the reason to exclude the data. The increase of diffuse light might be, but it is not a done deal either. Even in the case of increasing diffuse light, a better practice is not to exclude the data, but to include the variable (diffuse light) to the radiation time series included in the model.
6. Do we know what mechanism or mechanisms caused the loss of correlation between CGR and temperature in the mid-1970s? Was it caused by some variables not included in the model?
7. What happens if you exclude this period from the time series as well? This is to avoid potential critique on selective exclusion of data based on our (presumed) knowledge about the events.

I will recommend the publication of this paper upon receiving satisfactory responses from the authors on the above raised questions.

REVIEWER COMMENTS

Reviewer #1 (Remarks to the Author):

It seems that the authors still didn't understand my critique. So I will do it one last time and try my best to clarify my points. I will go back to my comments of the first round as there turned out to be a mistake in my second-round comments, which should be dismissed – and I apologize for it.

Let me start by saying that the *variations or anomalies of a physical variable (e.g., temperature) do not exist independently by themselves*. If we calculate anomalies without a clear idea of the physics, we could introduce errors. Therefore, we need to be extra careful whenever we do additional manipulations (including detrending, masking, etc.) to the data. Verify the results with the original simple time series as much as you can.

To illustrate this point, let us return to the original equation that describes the dynamics of atmospheric CO2 growth rate:

$$\dot{E} - \dot{C} = f(C, T, \dots) \quad (1)$$

where \dot{E} is the annual CO2 emission rate, \dot{C} is the annual CO2 growth rate, C is the atmospheric CO2 level, and T is temperature. Other variables such as precipitation (P) and radiation (R) are implicitly included in the dynamic system. $E - C$ represents the annual carbon sink and the physical meaning of the equation is very clear.

The function f is nonlinear and can be linearized around a point $X = (C_x, T_x, \dots)$

$$\dot{E} - \dot{C} = f(X) + \beta_x \cdot C'_x + \gamma_x \cdot T'_x + \dots \quad (2)$$

where the sensitivity coefficients are defined by the Jacobians at the point X . The term $f(X)$ is generally non-zero unless X is an equilibrium point of the system, such that

$$f(X) = 0 \quad (3)$$

We may assume that the climatology of variables at the beginning of the studying period (1950s-1960s) represents a quasi-equilibrium, such that

$$\dot{E} - \dot{C} = \beta_x \cdot C'_x + \gamma_x \cdot T'_x + \dots \quad (4)$$

(The climatologies in each of the 20 years moving window are not equilibrium, and therefore a constant term should be included in the regression equation. It is a better practice than removing the means of the variables because we can then check if the means satisfy the same equation.)

Next the paper “detrended” the time series. This is not a desirable practice, but if we do want to separate the long-term (LT) and the interannual (IA) variations of the time series, we must observe certain constraints. As described in my previous comment, we decompose the terms in Eq. (4) in to LT and IA components (the LT components are not necessarily a linear trend, but this is trivia and I won't distinguish the terms).

$$\dot{E}_{LT} - \dot{C}_{LT} - \dot{C}_{IA} = \beta \cdot C'_{LT} + (\gamma_{LT} + \gamma_{IA}) \cdot (T'_{LT} + T'_{IA}) + \dots \quad (5)$$

where we neglected the interannual component of CO2 emissions (\dot{E}_{IA}) and atmospheric CO2 concentration (C'_{IA}) for they are generally small. Because this study focuses on the changes of CGR's sensitivity on temperature (γ), we also decompose it as well. Now we can separate the long-term trend and the interannual variations as follows:

$$\dot{E}_{LT} - \dot{C}_{LT} = \beta \cdot C'_{LT} + \gamma_{LT} \cdot T'_{LT} + \dots \quad (6)$$

and

$$\dot{C}_{IA} = -(\gamma_{LT} + \gamma_{IA}) \cdot T'_{IA} + \gamma_{IA} \cdot T'_{LT} + \dots \quad (7)$$

The paper separates the two components in a different way as

$$\dot{E}_{LT} - \dot{C}_{LT} = \beta \cdot C'_{LT} + (\bar{\gamma} + \gamma') \cdot T'_{LT} + \dots \quad (6^*)$$

$$\dot{C}_{IA} = -(\bar{\gamma} + \gamma') \cdot T'_{IA} + \dots \quad (7^*)$$

This treatment is less optimal because it leaves the IA variations $\gamma_{IA} \cdot T'_{LT}$ unaccountable.)

It is clear that

- *The long-term component of TT'_{LT} is certainly not zero and it has important impacts on C'_{IA} with the sensitivity γ_{IA} .* This term is missing from the model used in the paper because the time series were detrended and, most likely, have the means removed.

(This responds to the authors' rebuttal that says "The temperature data (as well as other data) were detrended prior to use in the analysis, and in this process, the offset is removed, resulting in a time series that varies around 0... Hence, T_clt in the reviewer's equation is 0 by default, so the term disappears")

- More critically, because the paper detrended the data, Eq. (6) totally disappears in the analysis. We really need Eq. (6) to verify the results reported in the paper because both Eqs. (6) and (7) must be simultaneously satisfied so that Eq. 5 is satisfied. This verification is important because, again, the interannual variations or anomalies of a physical variable (e.g., temperature) are pretty much an artificial concept and do not exist independently by themselves.

Therefore, what I asked the authors to do in my first-round comments were:

1. Estimate $\bar{\gamma}$ and γ'' using Eq. (7);
2. Estimate β using Eq. (6) and the results ($\bar{\gamma}$ and γ'') from the preceding step. Check if the estimates of β (CO2 fertilization effect) comparable to those reported recently.
3. Predict C (i.e., $C_{\#} + C_{\&}$) using Eq. (5) and compare the results to the observations.

Finally, I also want to read more discussions on the exclusion of Pinatubo eruption era (1992-1993) from the data, in particular,

4. What will happen if you don't exclude this time period from the analysis?

5. I don't think extra cooling is the reason to exclude the data. The increase of diffuse light might be, but it is not a done deal either. Even in the case of increasing diffuse light, a better practice is not to exclude the data, but to include the variable (diffuse light) to the radiation time series included in the model.

6. Do we know what mechanism or mechanisms caused the loss of correlation between CGR and temperature in the mid-1970s? Was it caused by some variables not included in the model?

7. What happens if you exclude this period from the time series as well? This is to avoid potential critique on selective exclusion of data based on our (presumed) knowledge about the events.

I will recommend the publication of this paper upon receiving satisfactory responses from the authors on the above raised questions.

Response: We thank reviewer #1 very much for his/her patience, and we believe that we have fully understood the questions raised. We provide below a detailed explanation for the questions placed, the code for calculations and the data including all the relevant variables (with a step-by-step annotation).

We used annual mean temperature (TMP) during 2000-2019 (20 years corresponding to a moving window used in this study) as an example showing the course of the data processing. The red line gives the real observable TMP (K) and the black line is the linear regression line based on the least square method (Fig. 1a). We then can derive the interannual variations of TMP (real TMP subtracting its linear trend) as shown in Fig. 1b. The other relevant variables were processed in the same way to derive the two components of long term trend and interannual variations of time series (see data preparations in the code). We provide answers to the questions in the following:

Question 1, We ingest these processed variables into Eq (7) and show that the changes in Gamma ($G_{It} + G_{IA}$) are essentially the same as the changes in Gamma obtained by Eq (7*) used in our manuscript (in Fig. 1c), while G_{IA} based Eq (7) varies around 0 (Fig. 1d). The interception varies around 0 that can be neglected when calculating Beta based on Eq (6).

Question 2, Beta was estimated according Eq (6) and shows a reasonable change and in particular a decreasing trend in the past decades as reported (Fig. 1e) (Ref 1 below). Here data including CO₂ emission were extracted from the Global Carbon Project and the atmospheric CO₂ concentration from Global Monitoring Laboratory. We used Emission as the sum of emissions from fossil emissions excluding carbonation and land-use change emissions.

Question 3, The CGR was predicted for each moving window and agreed well with changes in observable CGR, with a significant ($p < 0.05$) correlation coefficient (r) of 0.87 (Fig. 1f). Note that the interceptions derived in Eq (6) should be subtracted in Eq (5) to predict CGR.

Question 4, We show the changes in Gamma over time including 1992-1993 in Fig. 1e, which is generally in line with Gamma without accounting for 1992-1993 (Fig. 1g), but with a peak in Gamma occurring around 1985 representing the 20-year window during year 1975-1995.

Question 5, Thanks for pointing this out. We have chosen to delete the text "in addition to global cooling" in the main part, and on the other hand added some relevant descriptions on the environmental impacts from Mt Pinatubo volcanic eruption in Line 368-372 (method section) supporting why data from this period was omitted: "This is because the Mt Pinatubo eruption was suggested to exert strong effects on the carbon cycle by increasing diffuse light that promotes photosynthesis²², as well as causing a decrease in net shortwave radiation that may constrain photosynthesis and causing a reduction in precipitation⁴³ and cooling⁴⁴ that both can affect photosynthesis and heterotrophic respiration."

We consider the exclusion of this particular period to be adequate as the eruption represents an anomaly that goes beyond what is considered normal climatic variability and it follows the approach taken in previous published papers on this particular subject (Ref 3 and 4 below) thereby making our results directly comparable with the current understanding of the subject.

Question 6, The decrease in the coupling between CGR and temperature in the mid-1970s can be primarily attributed to highly negative anomalies of precipitation following wet years. This is probably because low levels of precipitation cause water stress conditions (and legacy) for photosynthesis decreasing the carbon sink and thereby increasing the carbon emission dependency on given changes in temperatures.

Question 7, Thanks, from the answer to question 6 we hope the reviewer agree with us that these phenomena represent quite different conditions that should be dealt with in different ways. We have chosen to remove the years influenced by the volcanic eruption, on the one hand, as it represents an extraordinary disturbance that causes significant impact on carbon cycle (Ref 2) going beyond climate variability (it is a common practice in a large body of research using similar datasets by

removing such anomaly (Ref 3-4). On the other hand, the situation in the 1970s is a different one (see the response to question 6), as there is nothing that should lead us to suspect an anomaly in addition to climate events and even the quality of the data that may cause such decoupling of CGR and temperature.

Reference:

- 1, Wang, S. et al. (2020). Recent global decline of CO₂ fertilization effects on vegetation photosynthesis. *Science*, 370(6522), 1295–1300.
- 2: Sarmiento, J. L., et al. (2010). Trends and regional distributions of land and ocean carbon sinks. *Biogeosciences*, 7(8), 2351–2367.
- 3, Wang, X., et al. (2014). A two-fold increase of carbon cycle sensitivity to tropical temperature variations. *Nature*, 506(7487), 212–215.
- 4, Humphrey, V., et al. (2018). Sensitivity of atmospheric CO₂ growth rate to observed changes in terrestrial water storage. *Nature*, 560(7720), 628–631.

Fig.1 (a) Changes in mean annual temperature (TMP) during 2000-2019 (a 20-y moving window) and the trend of TMP; (b) interannual variability of TMP (TMP – TMP_trend); (c) Changes in Gamma ($G_{lt} + G_{ia}$) based Eq (7) and Gamma based on Eq (7*) over time including 1992-1993; (d) Changes in G_{IA} based on Eq (7); (e) Changes in Beta over time based on Eq(6); (f) Change in predicted CGR ($PgC\ y^{-1}$) based on Eq (5); (g) Changes in Gamma based on Eq (7) over time excluding 1992-1993.

R Code for calculation:

CGR and climate

```
f <- '/media/hbs333/wma/WM_2020/Data/CO2/revised/change_IN/CO2a2_series_CRU-12mon-  
window-1960-2020-nodetrend_annual.csv'  
dt <- read.table(file =f, header=T, sep = ",")  
nrow(dt)  
head(dt)
```

Emission and atmospheric CO₂ concentration

```
f <- '/media/hbs333/wmc/Land-C/CO2/carbon_budget.csv'  
dfst <- read.table(file =f, header=T, sep = ",")  
dfst$conc_co2 <- dfst$conc_co2*2.124  
dfst$emission <- dfst$fossil.emissions.excluding.carbonation + dfst$land.use.change.emissions  
dfst <- subset(dfst, Year >1959 & Year <2021)  
nrow(dfst1)
```

for 20 year moving windows

```
daf <- data.frame()  
cgrd <- array(-9999, dim=c(42, 20))  
nb <- nrow(dt)-19
```

loop statrts

```
for (w in 1:nb){  
  
  ### set moving window  
  samp <- dt[w:(w+19),]  ## including variables tmp  
  samp1 <- dfst[w:(w+19),] ## carbon emission and concentration
```

data preparations

```
dat1 <- data.frame(E_lt =samp1$emission- detrend(samp1$emission,'linear'),  
                  CGR_lt = samp1$co2 - detrend(samp1$co2,'linear'),  
                  Con_lt = samp1$conc_co2- detrend(samp1$conc_co2,'linear'), ##(CO2 concentration  
trend)  
                  CGR_ia = detrend(samp1$co2),  
                  tmp_ia = detrend(samp1$tmp,'linear'), tmp_lt = samp1$tmp- detrend(samp1$tmp,'linear'))
```

Question 1, calculate Gamma following Eq.7, using absolute sign here and should add a adverse sign in the following calculation

```
model_bootstrap <- lm(CGR_ia ~ tmp_ia + tmp_lt, data = dat1) ## Eq 7  
coef_x1 <- model_bootstrap$coefficients[2] ## (G_lt + G_IA)  
coef_x2 <- model_bootstrap$coefficients[3] ## (G_IA)  
const <- model_bootstrap$coefficients[1] ## intercepts varying around 0 can be neglected in Eq. 6
```

Question 2, calculate Beta using Eq.6

```
dfss <- data.frame(y = dat1$E_lt - dat1$CGR_lt + dat1$tmp_lt*(coef_x1- coef_x2), Con_lt =  
dat1$Con_lt)
```

```
fit <- lm(y ~ Con_lt, data = dfss)
beta <- fit$coefficients[2]
const <- fit$coefficients[1] ## Note intercepts should be kept and subtrated in Eq. 5
```

Question 3, predict CGR each windows (it will be a time series) using Eq. 5

```
cgr_pr <- dats$E_lt + samp$tmp*(coef_x1) - beta*dats$Con_lt - const
cgrd[w,] <- cgr_pr ## save per moving window
```

Gamma based on detrending variables as used in our manuscript

```
model_bootstrap <- lm(CGR_ia ~ tmp_ia, data = dats)
G2_ia <- model_bootstrap$coefficients[2]
```

```
## save data
```

```
dat <- data.frame(Gamma = coef_x1 ,G_ia = coef_x2, beta1 = beta, Ga_ia_dt = G2_ia)
daf <- rbind(daf,dat)
```

```
} ## end
```

```
daf$date <- 1970:2011
```

plot results

```
p1 <- ggplot(daf, aes(date, Gamma)) + geom_line(size=1.5,alpha=0.9, col= 'red') +
geom_line( aes(date,Ga_ia_dt))
```

```
p2 <- ggplot(daf, aes(date, beta1)) + geom_line(size=1.5,alpha=0.9)
```

```
p3 <- ggplot(daf, aes(date, G_ia)) + geom_line(size=1.5,alpha=0.9)
```

```
plot_grid(p1,p2,p3, ncol = 2, nrow = 2,labels = c('a','b','c'),align = 'hv')
```

REVIEWERS' COMMENTS

Reviewer #1 (Remarks to the Author):

The authors have correctly addressed my previous questions. The paper has reached the quality for publication.